# Scalable Learning from Probability Measures with Mean Measure Quantization

## Abstract

We consider statistical learning problems in which data are observed as a set of probability measures. Optimal transport (OT) is a popular tool to compare and manipulate such objects, but its computational cost becomes prohibitive when the measures have large support. We study a quantization-based approach in which all input measures are approximated by $K$-point discrete measures sharing a common support. We establish consistency of the resulting quantized measures. We further derive convergence guarantees for several OT-based downstream tasks computed from the quantized measures. Numerical experiments on synthetic and real datasets demonstrate that the proposed approach achieves performance comparable to individual quantization while substantially reducing runtime.

## 1 Introduction

Optimal transport (OT) (Villani, 2009; Peyré & Cuturi, 2019) provides a powerful framework to address modern machine learning problems involving collections of probability measures, this includes applications in signal and image processing, computer vision, and computational biology (Kolouri et al., 2017; Montesuma et al., 2024; Muandet et al., 2017; Khamis et al., 2024). As OT takes into account the geometry of the underlying space of the data, it has proven effective for computing averages of distributions through Wasserstein barycenters (Agueh & Carlier, 2011; Cuturi & Doucet, 2014), performing meaningful Principal Component Analysis (PCA) of probability measures (Wang et al., 2013; Seguy & Cuturi, 2015; Bigot et al., 2017; Cazelles et al., 2018), and several supervised learning tasks such as regression and classification . In particular, these problems are often addressed using linearized OT (LOT) (Wang et al., 2013; Delalande & Mérigot, 2023; Moosmüller & Cloninger, 2023), which allows to define an embedding of probability measures into a Hilbert space.

However, the computational cost of (L)OT-based methods makes them impractical when dealing with probability measures with large support. This is often the case when observing $N$ measures supported on point clouds with a large number $m_i$ of observations. Such datasets are frequently found in flow cytometry (McKinnon, 2018), where observations collected from $N$ patients represent point clouds of thousands to millions of cells (that is $m_i \geq 10^5$), each characterized by $d$ bio-markers, with $d$ larger than 10. Using OT-based learning methods on such raw data becomes computationally prohibitive as soon as the number $m_i$ of points per clouds exceeds a few thousands.

We then consider the problem of learning from $N$ large-support probability measures $\mu^{(1)}, \cdots, \mu^{(N)}$ on $\mathbb{R}^d$ using OT-based methods. A first natural solution to reduce computational costs is to approximate each distribution via random subsampling. Yet, this approach may fail to capture important geometric or low-density structures of the distributions. Quantization of probability measures (Graf & Luschgy, 2000; Pagès, 2015) provides an alternative, by approximating a measure by a discrete one with small support while controlling the Wasserstein error. One could therefore consider quantizing each input measure $\mu^{(i)}$ individually with $K$ points by solving the following $N$ quantization problems:

$$\min_{a^{(i)} \in \Sigma_K, \; X^{(i)} \in (\mathbb{R}^d)^K} W_2^2\left(\sum_{k=1}^K a_k^{(i)} \delta_{x_k^{(i)}}, \mu^{(i)}\right) \qquad \forall 1 \leq i \leq N, \tag{1}$$

where $X^{(i)}$ is the vector of points $x_1^{(i)}, \ldots, x_K^{(i)} \in \mathbb{R}^d$, and $\Sigma_K$ is the probability simplex in $\mathbb{R}^K$.

Problem 1 serves as a natural baseline; still, when the number $N$ of input measures is large, storing $N$ separate supports in $\mathbb{R}^d$ may be expensive. Furthermore, because each measure is quantized on its own support $\{x_k^{(i)}\}_{k=1}^K$, the resulting discrete approximations are not directly comparable across $i$. For instance, when the measures exhibit several subpopulations, the disappearance of one subpopulation within a subset of measures may be difficult to detect under individual quantization, since the centroids are not aligned across measures. In this paper, we therefore study an alternative quantization-based approach, in which all measures are approximated by discrete measures sharing a common support:

$$\min_{X \in (\mathbb{R}^d)^K} \frac{1}{N} \sum_{i=1}^N \min_{a^{(i)} \in \Sigma_K} W_2^2 \Big( \sum_{k=1}^K a_k^{(i)} \delta_{x_k}, \mu^{(i)} \Big). \tag{2}$$

Compared to Equation (1), Problem (2) yields a single support shared across all measures, while allowing measure-specific weights. This formulation reduces storage costs and provides a representation with a clear intepretation: all measures are expressed over a common set of atoms, so their weight vectors can be compared directly. In particular, when the measures contain multiple subpopulations, the disappearance of one subpopulation will translate to zero mass assigned to the corresponding centroid, see Figure 2. Formulation (2) mirrors common practice for data analysis involving multiple distributions: one fits a clustering model on the pooled data, and each distribution is then characterized by recovering proportional contributions for each cluster (Chazal et al., 2021; Naim et al., 2014).

## 1.1 Contributions

We study the shared-support quantization defined by Equation (2) and analyze its theoretical and practical implications for OT-based learning with collections of probability measures. We summarize our contributions as follows:

1. We highlight that the shared-support quantization problem (2) is equivalent to an optimal quantization of the mean measure in Proposition 3.1. Building on this equivalence, we establish consistency results for the resulting quantized measures. In particular, we prove convergence in the 2-Wasserstein distance between probability measures supported on the space of probability measures in Theorem 3.3.

2. As a consequence, we derive convergence guarantees for several downstream OT-based learning tasks computed from the quantized measures in Problem (2), including Wasserstein barycenters, statistical dispersion, and covariance operators associated with linearized OT embeddings in Section 4.

3. We illustrate the soundness and consistency of this quantization method through numerical experiments on synthetic and real datasets. In particular, we show that the mean measure used to construct the quantized measures can be subsampled by a factor $N$ without degrading performance. As a result, our shared-support quantization method (2) is computationally advantageous over individual quantization (1) in large-scale settings.

## 1.2 Related works

While quantization allows to approximate probability measures with a small set of points, other methods also aim at summarizing a dataset with representative samples. Within the framework of coresets (Huggins et al., 2016; Claici et al., 2018), one selects a subset of points such that solving a particular problem on this subset yields similar results than solving the problem on the entire dataset. In order to reduce the computational complexity of Gaussian Processes (GP) model, the principle of inducing variables, see e.g. Titsias (2009), also allows for an approximation of the posterior by choosing a set of representative points and conditioning the GP on these points.

In Chazal et al. (2021); Royer et al. (2021), quantization is employed to embed a set of $N$ probability measures into a finite-dimensional Euclidean space through measure vectorization. More precisely, given $N$ input measures $\mu^{(i)}$, a quantization of the mean measure $\bar{\mu} = \frac{1}{N} \sum_{i=1}^{n} \mu^{(i)}$ by $K$ centers $x_1, \ldots, x_K$ in $\mathbb{R}^d$ is first done. Then, each measure $\mu^{(i)}$ is mapped to $v^{(i)} = (v_1^{(i)}, \cdots, v_K^{(i)})$ a vector of the convex space $\mathbb{R}_+^K$, where $v_k^{(i)}$ roughly represents the mass of the measure $\mu^{(i)}$ distributed around the center $x_k$. Yet, this embedding does not take into account the relative positions of the $K$ centers, and consistency of the approach is not studied. Finally, the benefits of a preliminary quantization step have been studied in Beugnot et al. (2021) to improve the standard plug-in estimator of the OT cost between two probability measures.

## 2 Background

### 2.1 Optimal transport

Let $\rho$ and $\mu$ be two probability measures with support included in a compact set $\mathcal{X} \subset \mathbb{R}^d$. For the quadratic cost, the OT problem between $\rho$ and $\mu$ is:

$$W_2^2(\rho, \mu) := \min_{\pi \in \Pi(\rho, \mu)} \int_{\mathcal{X} \times \mathcal{X}} \|x - y\|^2 \mathrm{d}\pi(x, y), \tag{3}$$

where $\Pi(\rho, \mu)$ is the set of probability measures (or transport plans) on $\mathcal{X} \times \mathcal{X}$ with marginals $\rho$ and $\mu$. We denote by $\pi^*$ an optimal plan in Equation (3), and set $W_2(\rho, \mu) := \sqrt{W_2^2(\rho, \mu)}$. The OT problem also admits another formulation, called the Monge problem:

$$W_2^2(\rho, \mu) = \min_{T_{\#}\rho = \mu} \int_{\mathcal{X}} \|x - T(x)\|^2 \mathrm{d}\rho(x), \tag{4}$$

where the pushforward of a measure $\rho$ in $\mathbb{R}^d$ by a measurable map $T$ is defined as the measure $T_{\#}\rho$ such that for all Borelian $B \subset \mathbb{R}^d, T_{\#}\rho = \rho\left(T^{-1}(B)\right)$. When $\rho$ is absolutely continuous, we get from Brenier's theorem (Brenier, 1991)[Theorem 1.2] that (4) admits a unique minimizer, which we denote $T_\rho^\mu$. Furthermore, we have that the OT plan $\pi^*$ verifies $\pi^* = (\mathrm{id}, T_\rho^\mu)_{\#}\rho$.

Now, we endow the set of probability measures $\mathcal{P}(\mathcal{X})$ with the 2-Wasserstein distance $W_2$, which results in a curved space of probability measures. In this paper, we shall represent the set $(\mu^{(i)})_{1 \leq i \leq N}$ as the discrete (empirical) probability measure $\mathbb{P}^N = \frac{1}{N} \sum_{i=1}^{N} \delta_{\mu^{(i)}}$ over $\mathcal{P}(\mathcal{X})$. To define a metric on $\mathcal{P}(\mathcal{P}(\mathcal{X}))$, the set of Borel probability measures over $\mathcal{P}(\mathcal{X})$, we will use $W_2^2$ as the ground cost on the metric space $(\mathcal{P}(\mathcal{X}), W_2)$. The 2-Wasserstein distance over $\mathcal{P}(\mathcal{P}(\mathcal{X}))$ (Le Gouic & Loubes, 2017) is then defined as

$$\mathcal{W}_2(\mathbb{P}, \mathbb{Q}) = \left( \min_{\gamma \in \Gamma(\mathbb{P}, \mathbb{Q})} \int_{\mathcal{P}(\mathcal{X}) \times \mathcal{P}(\mathcal{X})} W_2^2(\rho, \mu) \mathrm{d}\gamma(\rho, \mu) \right)^{1/2}, \tag{5}$$

where $\Gamma(\mathbb{P}, \mathbb{Q})$ is the set of probability distributions on $\mathcal{P}(\mathcal{X}) \times \mathcal{P}(\mathcal{X})$ with respective marginals $\mathbb{P}$ and $\mathbb{Q}$.

### 2.2 Optimal quantization of probability measures

We briefly recall the notion of optimal quantization; see Graf & Luschgy (2000); Pagès (2015). Given a probability measure $\mu$ on $\mathbb{R}^d$ and an integer $K \geq 1$, the $K$-point quantization problem consists in approximating $\mu$ by a discrete probability measure supported on $K$ points, and is defined as

$$\min_{a \in \Sigma_K, \, X \in (\mathbb{R}^d)^K} W_2^2 \left( \mu, \sum_{k=1}^{K} a_k \delta_{x_k} \right), \tag{6}$$

where $\Sigma_K$ denotes the probability simplex in $\mathbb{R}^K$ and $X = (x_1, \ldots, x_K)$ with $x_k \in \mathbb{R}^d$. Any minimizer $(a^*, X^*)$ defines a $K$-point quantization of $\mu$, which is the measure $\sum_{k=1}^{K} a_k^* \delta_{x_k^*}$.

**Reformulation via Voronoï cells.** For $K \geq 1$ $X \in (\mathbb{R}^d)^K$, we have that

$$\min_{a \in \Sigma_K} W_2^2\left(\mu, \sum_{k=1}^K a_k \delta_{x_k}\right) = \int_{\mathbb{R}^d} \min_{1 \leq k \leq K} \|y - x_k\|^2 \mathrm{d}\mu(y), \tag{7}$$

see Lemma A.1. Moreover, if $\mu$ is absolutely continuous, a minimizer is given by $a_k^* = \mu(V_{x_k})$, where $(V_{x_k})_{k=1}^K$ are Voronoï cells defined as

$$V_{x_k} := \left\{ y \in \mathbb{R}^d \ : \ \|y - x_k\|^2 \leq \|y - x_\ell\|^2 \ \forall \ell \neq k \right\} \qquad \text{for } 1 \leq k \leq K. \tag{8}$$

These Voronoï cells form a partition of $\mathbb{R}^d$ up to a $\mu$-null set. When $\mu$ assigns positive mass to Voronoï boundaries (which may occur in particular for discrete measures), a measurable tie-breaking construction of a partition of $\mathbb{R}^d$ is required; see Equation (19) in Appendix A. Throughout the paper, we keep the notation $V_{x_k}$ for Voronoï cells, understanding it as the measurable partition produced by this tie-breaking when needed. As a consequence, the quantization Problem (6) rewrites as

$$\min_{X \in (\mathbb{R}^d)^K} \int_{\mathbb{R}^d} \min_{1 \leq k \leq K} \|y - x_k\|^2 \mathrm{d}\mu(y). \tag{9}$$

**Quantization error.** Given a minimizer $X^* \in (\mathbb{R}^d)^K$ of (9), the associated quantization error is defined by

$$\varepsilon_K(\mu) := \int_{\mathbb{R}^d} \min_{1 \leq k \leq K} \|y - x_k^*\|^2 \mathrm{d}\mu(y). \tag{10}$$

According to (Graf & Luschgy, 2000, Theorem 6.2), $\varepsilon_K(\mu) = \mathcal{O}(K^{-2/d})$ when $\mu$ is absolutely continuous, and $\varepsilon_K(\mu) = o(K^{-2/d})$ when $\mu$ is discrete.

## 3 Main results

Let $\mu^{(1)}, \cdots, \mu^{(N)}$ be $N$ probability measures in $\mathcal{P}(\mathcal{X})$ where $\mathcal{X} \subset \mathbb{R}^d$ is a compact set. To reduce computational costs when dealing with measures with large supports, we propose to approximate the input measures using a shared set of $K$ atoms and measure-specific weights. The method consists in solving the following problem:

$$\min_{a \in (\Sigma_K)^N} \min_{X \in (\mathbb{R}^d)^K} \frac{1}{N} \sum_{i=1}^N W_2^2\left(\sum_{k=1}^K a_k^{(i)} \delta_{x_k}, \mu^{(i)}\right), \tag{11}$$

as introduced in Gachon et al. (2025). The following result shows that Problem (11) is surprisingly equivalent to an optimal quantization problem of the mean measure $\overline{\mu} = \frac{1}{N} \sum_{i=1}^N \mu^{(i)}$.

**Proposition 3.1.** *Let $(\mu^{(i)})_{1 \leq i \leq N}$ be arbitrary probability measures with support included in a compact set $\mathcal{X} \subset \mathbb{R}^d$ and let $\overline{\mu} = \frac{1}{N} \sum_{i=1}^N \mu^{(i)}$ be the mean measure. Then,*

$$\min_{a \in (\Sigma_K)^N, \ X \in (\mathbb{R}^d)^K} \frac{1}{N} \sum_{i=1}^N W_2^2\left(\sum_{k=1}^K a_k^{(i)} \delta_{x_k}, \mu^{(i)}\right) = \min_{X \in (\mathbb{R}^d)^K} W_2^2\left(\sum_{k=1}^K \overline{\mu}(V_{x_k}) \delta_{x_k}, \overline{\mu}\right) \tag{12}$$

Proposition 3.1 is proven in Appendix C.1. For a minimizer $\bar{X} = (\bar{x}_1, \cdots, \bar{x}_K)$ of Equation (12), that is a $K$-point quantization of $\overline{\mu}$, we then define the quantized measures for $1 \leq i \leq N$ by

$$\mu_K^{(i)} = \sum_{k=1}^K \bar{a}_k^{(i)} \delta_{\bar{x}_k}, \ \text{with } \bar{a}_k^{(i)} = \mu^{(i)}(V_{\bar{x}_k}), \tag{13}$$

The measure $\mu_K^{(i)}$ is therefore a discrete probability measure supported on $K$ points which is an approximation of $\mu^{(i)}$ in the sense of the minimization problem in Equation (11). The measures $(\mu_K^{(i)})_{1 \leq i \leq N}$ differ in their

weights but share the same support $\overline{X}$. In a slight abuse of language, we will refer to $\mu_K^{(i)}$ as a quantized version of $\mu^{(i)}$, even though it is not a solution of the optimal quantization problem (6). We refer to the approximation of the $\mu^{(i)}$'s by the $\mu_K^{(i)}$'s as mean measure quantization.

*Remark* 3.2 (On the compactness assumption of $\mathcal{X}$). Proposition 3.1 remains true without the compactness assumption on $\mathcal{X}$, under finite 2-order moments of the measures.

**Practical construction.** In practice, each measure $\mu^{(i)}$ is observed through samples $X^{(i)} = (X_1^{(i)}, \cdots, X_{m_i}^{(i)}) \in (\mathbb{R}^d)^{m_i}$, $1 \leq i \leq N$, where $X_j^{(i)} \overset{iid}{\sim} \mu^{(i)}$. The mean measure $\overline{\mu}$ is then approximated by the empirical mean measure $(1/N) \sum_{i=1}^N \hat{\mu}^{(i)}$ where $\hat{\mu}^{(i)} = (1/m_i) \sum_{j=1}^{m_i} \delta_{X_j^{(i)}}$. One can think of the empirical mean measure as a concatenation of all samples $(X^{(i)})_{1 \leq i \leq N}$. We then compute a $K$-point quantization of the empirical mean measure using Lloyd's algorithm (Lloyd, 1982) with an initialization based on $K$-means++ (Arthur & Vassilvitskii, 2007). The time complexity of the Lloyd's algorithm is $O(Kd \sum_{i=1}^N m_i)$. This quantization yields centers $\bar{X} = (\bar{x}_1, \ldots, \bar{x}_K)$. Finally, for each $1 \leq i \leq N$, the weights of the quantized measure $\mu_K^{(i)}$ are computed by counting the number of points $X_j^{(i)}$ that fall into each Voronoï cell $V_{\bar{x}_k}$ for $1 \leq k \leq K$.

**Consistency of the mean measure quantization** The following result shows the consistency of the mean-measure quantization by leveraging the quantization error defined in Equation (10).

**Theorem 3.3.** *Let* $\mathbb{P}^N = \frac{1}{N} \sum_{i=1}^N \delta_{\mu^{(i)}}$, *and* $\mathbb{P}_K^N = \frac{1}{N} \sum_{i=1}^N \delta_{\mu_K^{(i)}}$ *where* $(\mu_K^{(i)})_{1 \leq i \leq N}$ *is given in Equation* (13). *Then,*

$$\mathcal{W}_2^2(\mathbb{P}_K^N, \mathbb{P}^N) = \frac{1}{N} \sum_{i=1}^N W_2^2(\mu^{(i)}, \mu_K^{(i)}) = \varepsilon_K(\overline{\mu}).$$

Theorem 3.3 is proven in Appendix C.2. Since $\mathcal{X}$ is a compact set, so is the metric space $(\mathcal{P}(\mathcal{X}), W_2)$ (Villani, 2009)[Remark 6.17]. Then, by Santambrogio (2015)[Theorem 5.9], $\mathcal{W}_2(\mathbb{P}_K^N, \mathbb{P}) \to 0$ if and only if $\mathbb{P}_K^N \to \mathbb{P}^N$ in the sense of weak convergence of distributions. In other words, for any bounded continuous function $f : \mathcal{P}(\mathcal{X}) \to \mathbb{R}$, it holds that $\int f(\nu) d\mathbb{P}_K^N(\nu) \overset{K \to +\infty}{\longrightarrow} \int f(\mu) d\mathbb{P}^N(\mu)$. Therefore, one can deduce from Theorem 3.3 the consistency of numerous statistics computed from the quantized measures $(\mu_K^{(i)})_{1 \leq i \leq N}$.

*Remark* 3.4. We prove an analogous version of Theorem 3.3 in Proposition B.1, when the $\mu_K^{(i)}$'s are constructed with individual quantization (1).

**Stability of the quantized measures.** Guaranteeing that the pairwise distance $W_2(\mu_K^{(i)}, \mu_K^{(j)})$ is a good approximation of $W_2(\mu^{(i)}, \mu^{(j)})$ is crucial for OT-based methods. Proposition 3.5, proven in Appendix D.3, provides such a guarantee when the input measures are all absolutely continuous.

**Proposition 3.5** (Pairwise distances). *Suppose that the probability measures* $(\mu^{(i)})_{i=1}^N$ *are absolutely continuous and that the support of the mean measure in included in* $[0,1]^d$. *Then, for any* $1 \leq i, j \leq N$,

$$W_2^2(\mu_K^{(i)}, \mu_K^{(j)}) \leq 3W_2^2(\mu^{(i)}, \mu^{(j)}) + \frac{6d}{\lfloor \sqrt[d]{K} \rfloor^2}.$$

## 4 Downstream tasks with quantized measures

We focus here on how downstream tasks computed from the quantized measures $\mu_K^{(i)}$ relate to the same tasks computed from the original measures $\mu^{(i)}$.

### 4.1 Wasserstein barycenter

A first example consists in proving that a Wasserstein barycenter (Agueh & Carlier, 2011) of the $(\mu_K^{(i)})_{1 \leq i \leq N}$ converges towards the unique Wasserstein barycenter of the measures $(\mu^{(i)})_{1 \leq i \leq N}$ when at least one of them is absolutely continous (a.c).

**Proposition 4.1** (Wasserstein barycenter). *Let $\mu_K^{\mathrm{bar}}$ be a Wasserstein barycenter of $(\mu_K^{(i)})_{1 \leq i \leq N}$ that is*

$$\mu_K^{\mathrm{bar}} \in \underset{\mu \in \mathcal{P}(\mathcal{X})}{\arg\min} \ \frac{1}{N} \sum_{i=1}^N W_2^2(\mu, \mu_K^{(i)}).$$

*If at least one of the measures $(\mu^{(i)})_{1 \leq i \leq N}$ is a.c., then $\mu_K^{\mathrm{bar}}$ converges to the unique Wasserstein barycenter $\mu^{\mathrm{bar}}$ of $(\mu^{(i)})_{1 \leq i \leq N}$ in the Wasserstein sense as $K \to +\infty$ .*

Proposition 4.1 is proven in Appendix D.1.

## 4.2 Statistical dispersion

We focus here on the convergence of the dispersion of the quantized measures towards the dispersion of the original measures.

**Proposition 4.2** (Statistical dispersion). *Let $\mathrm{SS}(\mu) = \frac{1}{N^2} \sum_{i,j=1}^N W_2^2(\mu^{(i)}, \mu^{(j)})$ and $\mathrm{SS}(\mu_K) = \frac{1}{N^2} \sum_{i,j=1}^N W_2^2(\mu_K^{(i)}, \mu_K^{(j)})$ be the statistical dispersion of $(\mu^{(i)})_{1 \leq i \leq N}$ and $(\mu_K^{(i)})_{1 \leq i \leq N}$ respectively. Then, for any $\lambda > 0$,*

$$\mathrm{SS}(\mu_K) \leq \big(1 + \frac{2}{\lambda}\big)\mathrm{SS}(\mu) + (4 + 2\lambda)\varepsilon_K(\overline{\mu}).$$

Proposition 4.2 shows that the dispersion of the quantized measures is controlled by the dispersion of the original measures and the quantization error. Proposition 4.2 is proven in Appendix D.2.

## 4.3 Clustering performances

We now show that the mean-measure quantization method preserve the clustering structure of the input measures. To this end, let us assume that each measure $\mu^{(i)}$ has a label $1 \leq l \leq L$. We note $I_l$ the set of indices such that $\forall i \in I_l, \mu^{(i)}$ has label $l$, and $N_l$ its cardinal. When clustering data, one usually aims at minimizing the within-class variance WCSS for a cluster $l$ and maximizing the between-class variance BCSS for clusters $l_1$ and $l_2$, where for a set of measure $\mu = (\mu^{(i)})_{i=1}^N$,

$$\mathrm{WCSS}(l, \mu) = \frac{1}{N_l^2} \sum_{i,j \in I_l} W_2^2(\mu^{(i)}, \mu^{(j)}), \qquad \mathrm{BCSS}(l_1, l_2, \mu) = \frac{1}{N_{l_1} N_{l_2}} \sum_{\substack{i_1 \in I_{l_1} \\ i_2 \in I_{l_2}}} W_2^2(\mu^{(i_1)}, \mu^{(i_2)}).$$

The next results, proven in Appendix D.4, gives a bound on the clustering performances of the quantized measures, as illustrated in Section 5.

**Proposition 4.3.** *For a given class $1 \leq l \leq L$, one has*

$$\mathrm{WCSS}(l, \mu_K) \leq 3\mathrm{WCSS}(l, \mu) + \frac{6N}{N_l}\varepsilon_K. \tag{14}$$

*For two distinct classes $l_1$ and $l_2$, one has that*

$$\mathrm{BCSS}(l_1, l_2, \mu_K) \geq \frac{1}{3}\mathrm{BCSS}(l_1, l_2, \mu) - \Big(\frac{N}{N_{l_1}} + \frac{N}{N_{l_2}}\Big)\varepsilon_K.$$

## 4.4 Application to Linearized OT embedding

In standard Euclidean settings, most ML methods, such as Principal Component Analysis (PCA) or Linear Discriminant Analysis (LDA), rely on the diagonalization of the covariance operator of the data. However, when the data belong to the space of probability measures endowed with the 2-Wasserstein distance, such methods cannot be applied directly as this space is not a Hilbert space. A common approach is to embed the probability measures into a Hilbert space. In this section, we focus on the linearized OT (LOT) embedding (Wang et al., 2013; Delalande & Mérigot, 2023; Moosmüller & Cloninger, 2023), which consists in the following:

$$\Phi^{\mathrm{LOT}} : \mathcal{P}(\mathcal{X}) \to L^2(\rho)$$
$$\mu \to T_\rho^\mu - \mathrm{Id}.$$

where $L^2(\rho) = \{v : \mathcal{X} \to \mathbb{R}^d \mid \int_{\mathcal{X}} \|v\|^2 \mathrm{d}\rho < \infty\}$ is endowed with the weighted $L^2$ inner product $\langle v_1, v_2 \rangle_{L^2(\rho)} = \int_{\mathcal{X}} v_1(x)^T v_2(x) \mathrm{d}\rho(x)$. We will assume that $\Phi^{\mathrm{LOT}}$ is centered, that is $\frac{1}{N} \sum_{i=1}^N \Phi^{\mathrm{LOT}}(\mu^{(i)}) = 0$.

We denote $\Sigma^N$ and $\Sigma_K^N$ the covariance operators associated with the embeddings of the original measures and the quantized measures respectively, that is

$$\Sigma^N = \frac{1}{N} \sum_{i=1}^N \Phi^{\mathrm{LOT}}(\mu^{(i)}) \otimes \Phi^{\mathrm{LOT}}(\mu^{(i)}) \qquad \Sigma_K^N = \frac{1}{N} \sum_{i=1}^N \Phi^{\mathrm{LOT}}(\mu_K^{(i)}) \otimes \Phi^{\mathrm{LOT}}(\mu_K^{(i)}). \tag{15}$$

We denote $\| \cdot \|_{\mathrm{HS}}$ the Hilbert-Schmidt norm on the space of Hilbert-Schmidt operators on $\mathcal{H}$. As $\Sigma^N$ and $\Sigma_K^N$ have finite rank (at most $N$), they are Hilbert-Schmidt operators and their Hilbert-Schmidt norms are therefore well defined.

**Proposition 4.4** (Covariance operator). *Let $\Sigma^N$ and $\Sigma_K^N$ be the covariance operators associated with the LOT embeddings of the original measures and the quantized measures respectively. Assume the Monge maps $T_\rho^{\mu^{(i)}}$ are $L$-Lipschitz for all $1 \leq i \leq N$. Then,*

$$\|\Sigma^N - \Sigma_K^N\|_{\mathrm{HS}} \leq 4\mathrm{diam}(\mathcal{X})^{3/2} L^{1/2} \, \varepsilon_K^{1/4}(\overline{\mu}).$$

Proposition 4.4 is proven in Appendix D.5. As a consequence of Proposition 4.4, we have that any ML task relying on the covariance operator $\Sigma^N$ is consistent (as $K \to +\infty$) with the same task computed from $\Sigma_K^N$.

*Remark* 4.5. In Appendix E, we derive an analogous result to Proposition 4.4 for the $N \times N$ Gram matrices $G^N$ and $G_K^N$ constructed such that for all $1 \leq i, j \leq N$,

$$(G^N)_{ij} = \langle \Phi^{\mathrm{LOT}}(\mu^{(i)}), \Phi^{\mathrm{LOT}}(\mu^{(j)}) \rangle_{L^2(\rho)} \quad \text{and} \quad (G_K^N)_{ij} = \langle \Phi^{\mathrm{LOT}}(\mu_K^{(i)}), \Phi^{\mathrm{LOT}}(\mu_K^{(j)}) \rangle_{L^2(\rho)}.$$

*Remark* 4.6. We derive a similar result to Proposition 4.4 for the Kernel Mean Embedding (KME) (Muandet et al., 2017) in Proposition D.4 in Appendix D.6. The KME is also a Hilbert space embedding of probability measures, which depends on the choice of a kernel function $k$. However, since our analysis revolves around optimal transport and the Wasserstein geometry, we chose to solely focus on the LOT embedding.

*Remark* 4.7. While the assumption that the Monge maps $T_\rho^{\mu^{(i)}}$ are $L$-Lipschitz in Proposition 4.4 can be restrictive, as it requires regularity assumptions on the measures $\rho$ and $\mu^{(i)}$, we adopt this formulation for simplicity. Alternative stability results (Merigot, 2026) for optimal transport maps are available under weaker assumptions and would lead different exponents in the inequality.

## 5 Numerical experiments

### 5.1 Synthetic datasets

#### 5.1.1 Gaussian measures

We illustrate the proposed mean-measure quantization scheme on a controlled setting in which several OT-based quantities can be computed in closed form. We consider $N = 20$ isotropic Gaussian measures on $\mathbb{R}^d$ ($d = 2$) of the form $\mu^{(i)} = \mathcal{N}\big(b^{(i)}, (\sigma^{(i)})^2 I_d\big), 1 \leq i \leq N$, where $b^{(i)} \in \mathbb{R}^d$ and $\sigma^{(i)} > 0$ are fixed. The Wasserstein barycenter of $(\mu^{(i)})_{i=1}^N$ admits a closed form (Agueh & Carlier, 2011):

$$\mu^{\mathrm{bar}} = \arg\min_{\mu \in \mathcal{P}(\mathbb{R}^d)} \frac{1}{N} \sum_{i=1}^N W_2^2(\mu, \mu^{(i)}) = \mathcal{N}\left(\frac{1}{N} \sum_{i=1}^N b^{(i)}, \left(\frac{1}{N} \sum_{i=1}^N \sigma^{(i)}\right)^2 I_d\right) \tag{16}$$

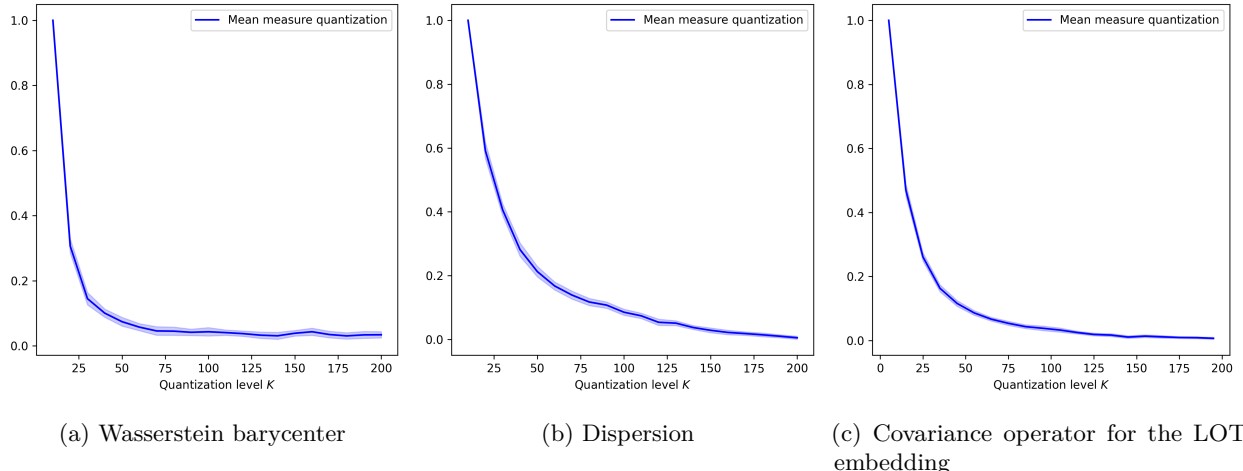

(a) Wasserstein barycenter

(b) Dispersion

(c) Covariance operator for the LOT embedding

Figure 1: Convergence of downstream quantities for the Gaussian synthetic dataset. Results are averaged over 20 independent trials and reported with 95% confidence interval.

For any $1 \leq i, j \leq N$, the squared 2-Wasserstein distance also admits the explicit expression

$$W_2^2\big(\mu^{(i)}, \mu^{(j)}\big) = \|b^{(i)} - b^{(j)}\|^2 + d\big(\sigma^{(i)} - \sigma^{(j)}\big)^2.$$

In particular, this allows us to compute exactly the statistical dispersion $\mathrm{SS}(\mu)$ defined in Proposition 4.2:

$$\mathrm{SS}(\mu) = \frac{1}{N^2} \sum_{i,j=1}^{N} \|b^{(i)} - b^{(j)}\|^2 + d\big(\sigma^{(i)} - \sigma^{(j)}\big)^2. \tag{17}$$

Finally, choosing the reference measure $\rho = \mathcal{N}(0, I_d)$, the Monge map from $\rho$ to $\mu^{(i)}$ is affine and the LOT embedding admits the explicit form

$$\Phi^{\mathrm{LOT}}\big(\mu^{(i)}\big)(x) = T_\rho^{\mu^{(i)}}(x) - x = (\sigma^{(i)} - 1)x + b^{(i)}.$$

This provides a closed-form ground truth for the associated covariance operator $\Sigma^N$ defined in Equation (15).

In practice, for each $1 \leq i \leq N$ we sample $m = 200$ points $X_1^{(i)}, \ldots, X_m^{(i)} \overset{iid}{\sim} \mu^{(i)}$ and define the empirical measures $\hat{\mu}^{(i)} = \frac{1}{m} \sum_{j=1}^{m} \delta_{X_j^{(i)}}$. For a range of values of $K$, we compute the quantized measures $\hat{\mu}_K^{(i)}$ by applying the practical construction described after Equation (13): we quantize the empirical mean measure and then define the weights by Voronoï counting. We then empirically evaluate the consistency of these quantities. In Figure 1a, we report $W_2^2\big(\mu_K^{\mathrm{bar}}, \mu^{\mathrm{bar}}\big)$, where $\mu^{\mathrm{bar}}$ is the closed-form barycenter in Equation (16) and $\mu_K^{\mathrm{bar}}$ is a Wasserstein barycenter of $(\hat{\mu}_K^{(i)})_{i=1}^N$. In Figure 1b, we plot $|\mathrm{SS}(\hat{\mu}_K) - \mathrm{SS}(\mu)|$, where $\mathrm{SS}(\hat{\mu}_K)$ is computed from $(\hat{\mu}_K^{(i)})_{i=1}^N$ and $\mathrm{SS}(\mu)$ is defined in Equation (17). To approximate the LOT embedding, we draw an empirical reference measure $\hat{\rho} = (1/m_0) \sum_{j=1}^{m_0} \delta_{y_j}$ with $m_0 = 100$ and $y_j \overset{iid}{\sim} \rho$. For each $i$, we solve the discrete OT problem between $\hat{\rho}$ and $\hat{\mu}_K^{(i)}$, yielding an optimal plan $P^{(i)} \in \mathbb{R}^{m_0 \times K}$, and we define the barycentric projection map

$$T_K^{(i)}(y_j) = m_0 \sum_{k=1}^{K} P_{jk}^{(i)} \bar{x}_k, \tag{18}$$

where $(\bar{x}_k)_{k=1}^K$ are the shared atoms constructed with the mean-measure quantization. We then compute the empirical covariance operator $\Sigma_K^N$ from the embeddings $\big(T_K^{(i)} - \mathrm{id}\big)$, and Figure 1c reports its Hilbert–Schmidt distance to the closed-form operator $\Sigma^N$.

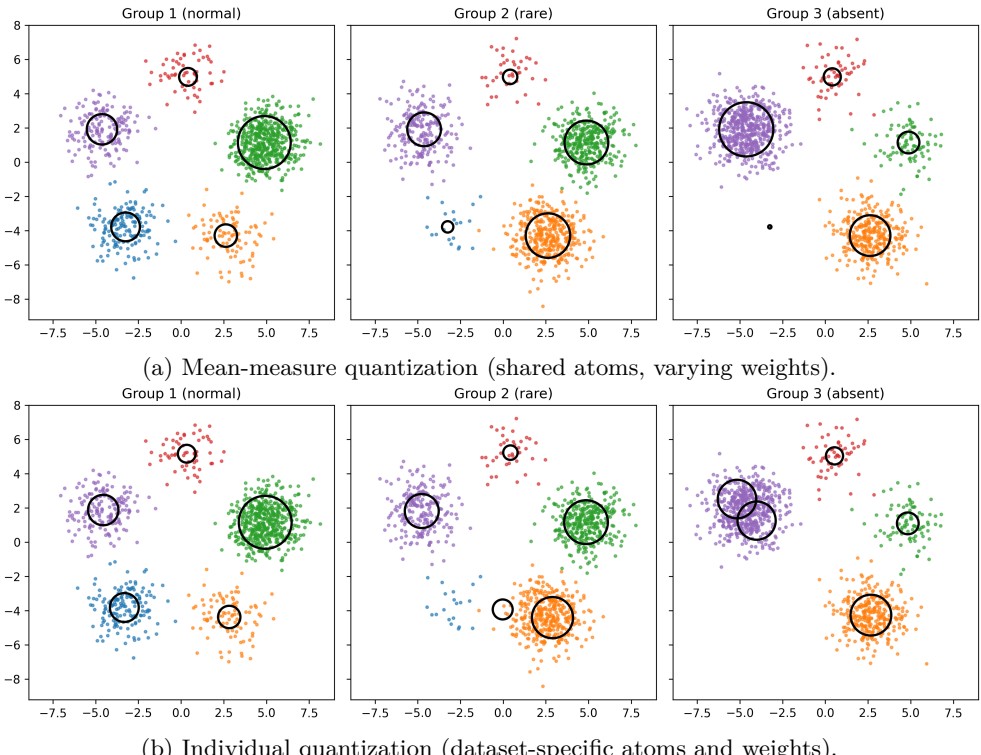

(a) Mean-measure quantization (shared atoms, varying weights).

(b) Individual quantization (dataset-specific atoms and weights).

Figure 2: Rare-population synthetic dataset: comparison of mean-measure quantization and individual quantization with the same number of centers ($K = 5$). The black circles represent the centers, and their radii are proportional to the corresponding weights.

### 5.1.2 Rare populations

This experiment highlights the benefit of mean-measure quantization when a small cluster is present only in a subset of the measures. We generate a collection of three measures on $\mathbb{R}^2$ as 5-component Gaussian mixtures with fixed locations and covariances. Across the three measures, only the mixture weights (i.e. vectors in the 5-dimensional simplex $\Sigma_5$) vary. One designated Gaussian component plays the role of a rare population whose weight is either comparable to the other components (Group 1), very small (Group 2), or 0 (absent) (Group 3). We then sample 1000 observations from each measure. The observations sampled from the Gaussian component illustrating the rare population are drawn in blue in Figure 2.

We compare the two quantization strategies with the same number of centers $K = 5$. Individual quantization (see Appendix B) runs a separate $k$-means per measure, yielding measure-specific atoms and weights. Mean-measure quantization runs a single global $k$-means on the pooled samples, yielding shared atoms across measures; only the weights vary from one measure to another.

Figure 2 visualizes the atoms and associated weights obtained with both methods. Note that in this experiment, the pooled samples contain observations from all three groups. With mean-measure quantization (top panel), measures are directly comparable since they share the same support, and the rare population manifests as a weight that vanishes (or becomes very small) in the corresponding group. With individual quantization (bottom panel), the atoms are not aligned across datasets, making the representation less interpretable.

### 5.2 Real datasets

In these real-world experiments, we focus on classification tasks computed from LOT embeddings of $K$-point discrete representations of the input measures. The goal is to reduce computational cost while preserving the information relevant for downstream learning. We compare the following approaches:

- **Mean-measure quantization (Full MMQ).** We compute our proposed shared-support quantization from the mean measure $\overline{\mu}$, yielding a support $(x_k)_{k=1}^K$ shared across all measures. Then, the quantized measures are $\sum_{k=1}^K \mu^{(i)}(V_{x_k})\delta_{x_k}$. For $N$ discrete measures $\mu_1, \cdots, \mu_N$ supported on $m_1, \cdots, m_N$ points respectively, the overall complexity of mean measure quantization followed by the LOT embedding of the quantized measures is $O(Kd\sum_{i=1}^N m_i) + O(NKm_0 \log(K + m_0))$, where the first term corresponds to the $k$-means step (up to a constant number of Lloyd iterations) and the second to solving the $N$ discrete OT problems (between the reference measure of size $m_0$ and the quantized measures of size $K$) involved in the LOT embedding.

- **Subsampled mean-measure quantization (Subsampled MMQ).** In Full MMQ, the mean measure $\overline{\mu}$ corresponds to a pooled dataset with $\sum_{i=1}^N m_i$ points, which can be very large. To reduce the cost of the $k$-means step, we subsample $\overline{\mu}$ by drawing $\frac{1}{N}\sum_{i=1}^N m_i$ points i.i.d. from $\overline{\mu}$, compute the shared atoms $(x_k)_{k=1}^K$ from this subsample, and then construct, for each $i$, the quantized measure $\sum_{k=1}^K \mu^{(i)}(V_{x_k})\delta_{x_k}$. The overall complexity of subsampled mean-measure quantization followed by the LOT embedding of the quantized measures is then $O\left(\frac{Kd}{N}\sum_{i=1}^N m_i\right) + O\left(NKm_0\log(K + m_0)\right)$.

- **Individual quantization (IQ) (Beugnot et al., 2021).** With individual quantization (1), we run a separate $k$-means per measure, resulting in measure-specific atoms and weights $\sum_{k=1}^K \mu^{(i)}(V_{x_k^{(i)}})\delta_{x_k^{(i)}}$.

- **Random subset (Rand).** We randomly sample $K$ points $(\tilde{x}_1^{(i)}, \cdots, \tilde{x}_K^{(i)})$ from $\mu^{(i)}$ and construct the associated (non-optimal) discrete measure $\sum_{k=1}^K \mu^{(i)}\left(V_{\tilde{x}_k^{(i)}}\right)\delta_{\tilde{x}_k^{(i)}}$.

- **Empirical measure (Emp).** We randomly select $K$ points $(\tilde{x}_1^{(i)}, \cdots, \tilde{x}_K^{(i)})$ from $\mu^{(i)}$ and construct the empirical measure $\frac{1}{K}\sum_{k=1}^K \delta_{\tilde{x}_k^{(i)}}$.

In the expressions above, $V_x$ denotes the (tie-broken) Voronoï cell associated with the support point $x$.

*Remark* 5.1. The consistency results derived in the previous sections apply to MMQ constructed from the full mean measure. In the numerical experiments, we also consider a version of MMQ where the empirical mean measure is approximated by a subsample before quantization. This additional subsampling step introduces an approximation error which is not considered in the current theoretical analysis.

### 5.3 Real dataset: flow cytometry

In this section, we use flow cytometry datasets provided in Thrun et al. (2022) and publicly available in Mendeley Data to illustrate the practical relevance of our method through a supervised classification task. The dataset is made of $N = 108$ cytometry measures (i.e. point clouds), corresponding to peripheral blood (pB), healthy bone marrow (BM) and leukemic bone marrow samples. Each measure constains from 100,000 to 1,000,000 points in dimension $d = 10$, which prevents the direct use of OT-based methods without a prior quantization step. For each method described above, we compute the LOT embeddings of the resulting $K$-point discrete representations. We use as reference measure the uniform measure on $[0,1]^d$, from which we sample $m_0 = 100$ points. The transport maps are approximated via barycentric projection (18) of the optimal transport plans between the empirical reference measure and the quantized measures. The embeddings lie in the linear space $\mathbb{R}^{K \cdot m_0}$, where we perform supervised classification to predict whether a sample corresponds to peripheral blood, healthy bone marrow, or leukemic bone marrow. We evaluate classification performance using stratified shuffle-split cross-validation, which preserves class proportions across splits. For each of 30 random splits (67% training and 33% testing), we fit an LDA classifier on the training set and report its accuracy on the test set. Figure 3a displays the mean classification accuracy together with its standard deviation for the different methods. Figure 3b reports the computational time required to compute the $K$-point discrete representations for each method. We find that the quantization-based approaches (MMQ and IQ) outperform the other methods. Interestingly, the classifier achieves essentially the same performance when using MMQ computed on the full dataset as when using subsampled MMQ. While individual quantization yields the highest accuracy results, subsampled MMQ remains very close in performance and offers a substantial reduction in computational cost.

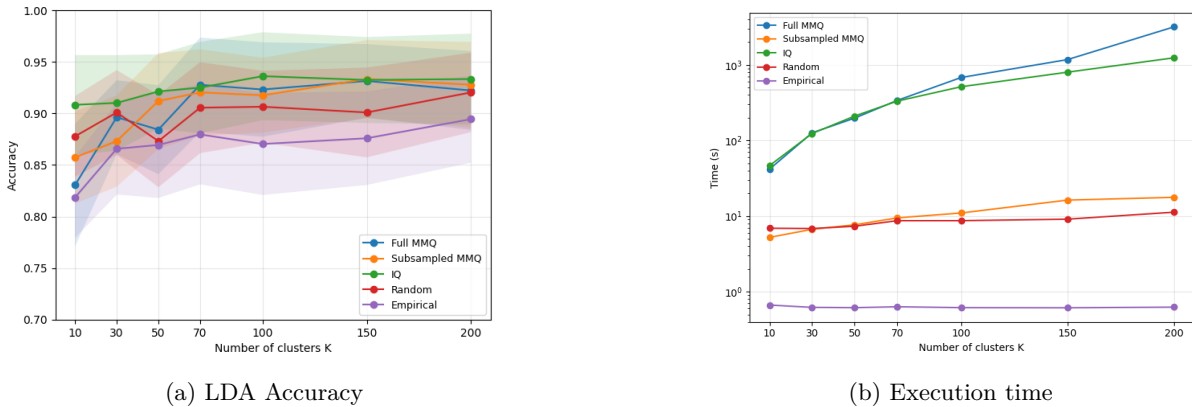

(a) LDA Accuracy

(b) Execution time

Figure 3: **Flow cytometry dataset.** LDA classification accuracy and executions times against $K$.

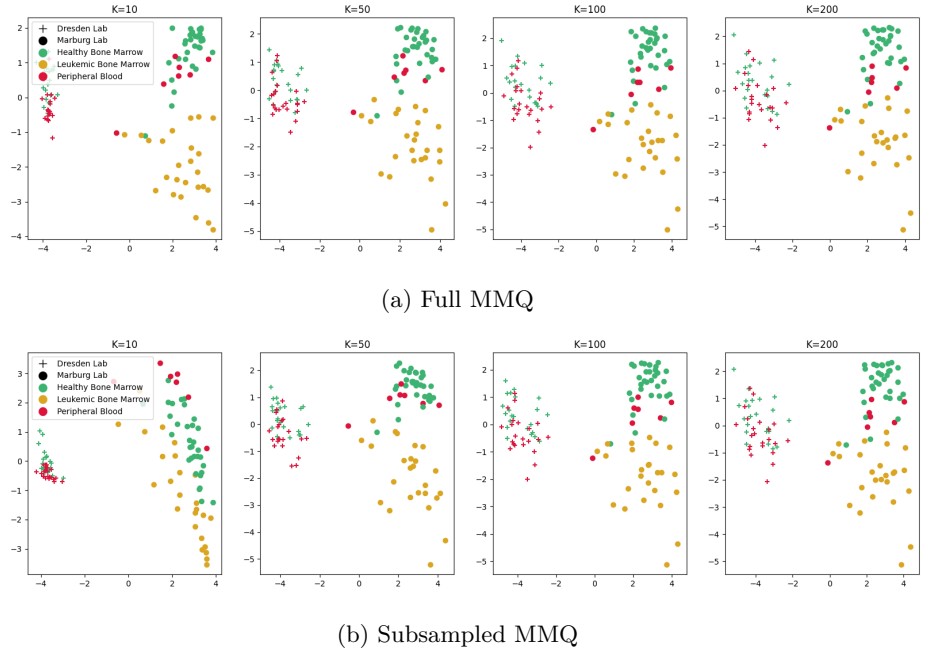

(a) Full MMQ

(b) Subsampled MMQ

Figure 4: **Flow cytometry dataset.** Projection of the $N = 108$ measures on the first two components of PCA.

Additionally, after computing the LOT embeddings of the quantized measures, we perform a 2-component PCA on the embedded measures and observe the projections of the data in Figure 4. It is clear that the representations stabilize when $K \geq 50$ and that large values of $K$ are not necessary.

### 5.4 Real dataset: earth images

We perform similar experiments on a set of images provided by the Airbus company. The dataset consists in $N = 1000$ images of size $128 \times 128$ captured by a SPOT satellite. The images are divided into two categories: those with the presence of a wind turbine and those without, see Figures 5a and 5b. Each image is viewed as a discrete probability distribution on the RGB space, that is each pixel is represented by a point in $\mathbb{R}^3$. The size $N$ of the dataset as well as the number of pixels ($m = 128^2$) prevents from directly using OT-based methods. We therefore reduce the size of supports with the different quantization methods for different

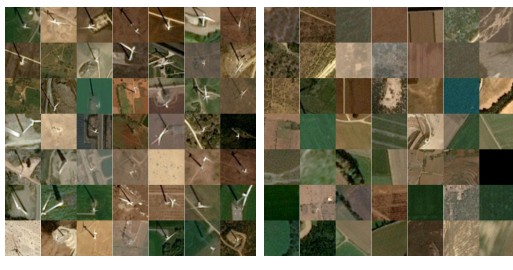

(a) With wind turbine    (b) Without wind turbines

Figure 5: **Earth image dataset.** Examples of images sampled from the Airbus dataset.

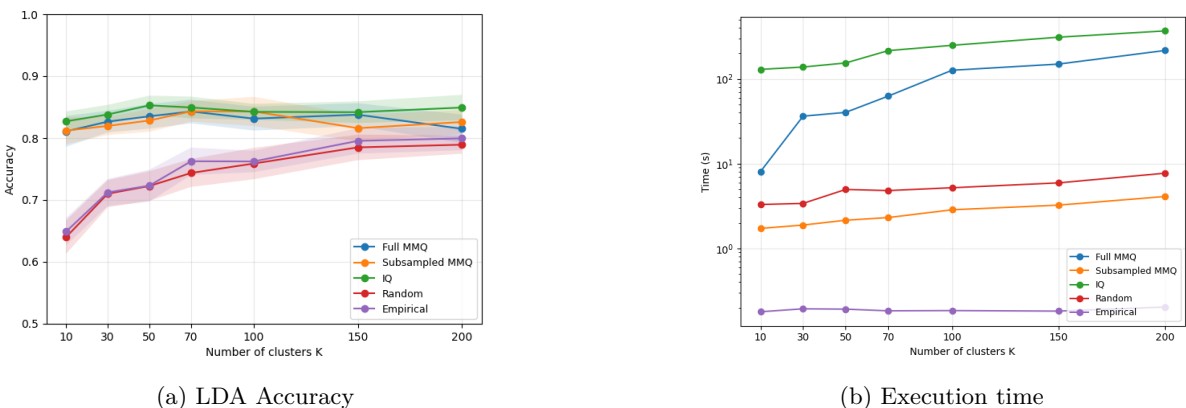

(a) LDA Accuracy                               (b) Execution time

Figure 6: **Earth image dataset.** LDA classification accuracy and execution times against $K$.

values of $K$. We then compute the LOT embeddings of the quantized measures, using as reference measure the uniform measure on $[0, 1]^3$, from which we sample $m_0 = 100$ points. We then carry out supervised classification. We also evaluate classification performance using stratified shuffle-split cross-validation. For each of 30 random splits (67% training and 33% testing), we fit an LDA classifier on the training set and report its accuracy on the test set. Figure 6a displays the mean classification accuracy together with its standard deviation for the different methods. Figure 6b reports the computational time required to compute the $K$-point discrete representations for each method. Once again, individual quantization yields better accuracy results, closely followed by both MMQ-based methods. Overall, subsampled MMQ offers the best trade-off between predictive performance and computational cost.

## 6 Conclusion

In this work, we studied mean-measure quantization, which approximates each input measure with a $K$-point discrete representation while enforcing a shared support across all measures. We established consistency results for the resulting quantized measures, as well as for several downstream tasks computed from them. Numerical experiments on synthetic and real datasets suggest that mean-measure quantization can be more efficient than individual quantization in large-scale settings: the large-support mean measure can be subsampled by a factor $N$ (number of input measures) with essentially no loss in performance, yielding substantial computational savings.

One limitation of our quantization approach is its sensitivity to the curse of dimensionality, a challenge common to many statistical problems in optimal transport. In future works, one could bypass the dependence on the ambient dimension by relying on a notion of intrinsic dimension, as studied in Weed & Bach (2019). Replacing the k-means objective with a projection-based alternative inspired by Sliced Optimal Transport could provide another route for avoiding dimensionality issues.

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

# A   Voronoï reformulation

Recall the classical Voronoï sets:

$$V_{x_k} = \left\{ y \in \mathbb{R}^d \| y - x_k \|^2 \leq \| y - x_\ell \|^2 \ \forall \ell \neq k \right\} \qquad \text{for } 1 \leq k \leq K.$$

For discrete probability measures, these sets may overlap on boundaries. Define inductively the following tie-breaking construction of Voronoï cells:

$$U_{x_1} = V_{x_1} \text{ and } U_{x_k} = V_{x_k} \backslash \bigcup_{j<k} U_{x_j} \qquad \text{for } k \geq 2, \tag{19}$$

Then $(U_{x_k})_{k=1}^K$ form a partition of $\mathbb{R}^d$ for $X \in (\mathbb{R}^d)^K$.

**Lemma A.1.** *Let $X = (x_1, \cdots, x_K) \in (\mathbb{R}^d)^K$ and consider the tie-breaking Voronoï partition defined in Equation (19). Then, for any probability measure $\mu \in \mathcal{P}(\mathcal{X})$, one has that*

$$\min_{a \in \Sigma_K} W_2^2 \Big( \mu, \sum_{k=1}^K a_k \delta_{x_k} \Big) = W_2^2 \Big( \mu, \sum_{k=1}^K \mu(U_{x_k}) \delta_{x_k} \Big)$$

$$= \sum_{k=1}^K \int_{U_{x_k}} \| x_k - y \|^2 \mathrm{d}\mu(y)$$

$$= \int_{\mathbb{R}^d} \min_{1 \leq k \leq K} \{ \| x_k - y \|^2 \} \mathrm{d}\mu(y).$$

*Proof of Lemma A.1.* Let $X = (x_1, \cdots, x_K) \in (\mathbb{R}^d)^K$. From the dual formulation of the Kantorovich problem (Peyré & Cuturi, 2019)[Equation 5.7], we have that, for any $a \in \Sigma_K$,

$$W_2^2 \Big( \mu, \sum_{k=1}^K a_k \delta_{x_k} \Big) = \sup_{\beta \in \mathbb{R}^K} \sum_{k=1}^K \beta_k a_k + \int_{\mathbb{R}^d} \beta^c(y) \mathrm{d}\mu(y)$$

$$= \sup_{\beta \in \mathbb{R}^K} \sum_{k=1}^K a_k \beta_k + \int_{\mathbb{R}^d} \left( \min_{1 \leq k \leq K} \{ \| x_k - y \|^2 - \beta_k \} \right) \mathrm{d}\mu(y)$$

$$\geq \int_{\mathbb{R}^d} \min_{1 \leq k \leq K} \{ \| x_k - y \|^2 \} \mathrm{d}\mu(y), \tag{20}$$

where the above inequality is obtain by taking $\beta_k = 0$ for all $1 \leq k \leq K$. Then, since $(U_{x_k})_{1 \leq k \leq K}$ is a partition of $\mathbb{R}^d$, and we may define

$$T_K(y) = \sum_{k=1}^K x_k \mathbf{1}_{U_{x_k}}(y) \text{ for } y \in \mathbb{R}^d,$$

that is a mapping from $\mathbb{R}^d$ to $X$. Introducing the probability measure $\mu_K = \sum_{k=1}^K \mu(U_{x_k}) \delta_{x_k}$, it is not difficult to see that $T_{K\#}\mu = \mu_K$ where the notation $T_{\#}\mu$ denotes the push-forward of a measure $\mu$ by the mapping $T$. Now, we let $\pi_K = (\mathrm{id} \times T_K)_{\#}\mu$ that obviously belongs to the set of transport plans $\Pi(\mu, \mu_K)$. From the definition of the Voronoï partition $(U_{x_k})_{1 \leq k \leq K}$, one can than check that, for any $y \in \mathbb{R}^d$, (see e.g. Pagès (2015))

$$\| y - T_K(y) \|^2 = \min_{1 \leq k \leq K} \{ \| x_k - y \|^2 \}.$$

Consequently, we obtain the following equalities

$$\int_{\mathcal{X}\times\mathcal{X}} \|x-y\|^2 \mathrm{d}\pi_K(x,y) = \int_{\mathbb{R}^d} \|y - T_K(y)\|^2 \mathrm{d}\mu(y) = \int_{\mathbb{R}^d} \min_{1\le k\le K}\{\|x_k - y\|^2\}\mathrm{d}\mu(y).$$

Inserting the above equality into (20), we thus have that, for any $a \in \Sigma_K$,

$$W_2^2\left(\mu, \sum_{k=1}^K a_k \delta_{x_k}\right) \ge \int_{\mathcal{X}\times\mathcal{X}} \|x-y\|^2 \mathrm{d}\pi_K(x,y).$$

Since $W_2^2\left(\mu, \mu_K\right) = \min_{\pi\in\Pi(\mu,\mu_K)} \int_{\mathcal{X}\times\mathcal{X}} \|x-y\|^2 \mathrm{d}\pi(x,y)$, we directly obtain from the above inequality that $W_2^2(\mu,\mu_K) = \int_{\mathcal{X}\times\mathcal{X}} \|x-y\|^2 \mathrm{d}\pi_K(x,y)$, which implies

$$\min_{a\in\Sigma_K} W_2^2\left(\mu, \sum_{k=1}^K a_k \delta_{x_k}\right) = W_2^2\left(\mu, \mu_K\right) = \int_{\mathbb{R}^d} \|y - T_K(y)\|^2 \mathrm{d}\mu(y) = \int_{\mathbb{R}^d} \min_{1\le k\le K}\{\|x_k - y\|^2\}\mathrm{d}\mu(y),$$

which concludes the proof. □

## B   Individual quantization of measures

When dealing with $N$ large-support measures $\mu^{(1)}, \cdots, \mu^{(N)}$, a natural approach is to independently approximate each measure $\mu^{(i)}$ by its optimal quantization:

$$\mu_K^{(i)} = \sum_{k=1}^K a_k^{(i)} \delta_{x_k^{(i)}},$$

where $a^{(i)}$ and $X^{(i)} = (x_1^{(i)}, \cdots, x_K^{(i)})$ are minimizers of (6) for $\mu = \mu^{(i)}$. Then, one can derive an analogue of Theorem 3.3 for individual quantization.

**Proposition B.1.** *Let* $\mathbb{P}^N = \frac{1}{N}\sum_{i=1}^N \delta_{\mu^{(i)}}$ *and* $\mathbb{P}_K^N = \frac{1}{N}\sum_{i=1}^N \delta_{\mu_K^{(i)}}$ *where* $(\mu_K^{(i)})_{1\le i\le N}$ *is the optimal quantization of* $(\mu^{(i)})_{1\le i\le N}$. *Then,*

$$\mathcal{W}_2^2(\mathbb{P}_K^N, \mathbb{P}^N) = \frac{1}{N}\sum_{i=1}^N W_2^2(\mu^{(i)}, \mu_K^{(i)}) = \frac{1}{N}\sum_{i=1}^N \varepsilon_K(\mu^{(i)}).$$

*Proof of Proposition B.1.* For a fixed $N \ge 1$, since $\mathbb{P}^N$ and $\mathbb{P}_K^N$ are discrete uniform measures of the same size, one can actually restrict the optimization set in (5) to permutations $\sigma \in \text{Perm}(N)$ (Peyré & Cuturi, 2019)[Equation 2.2] in the following sense:

$$\mathcal{W}_2^2(\mathbb{P}^N, \mathbb{P}_K^N) = \min_{\sigma\in\text{Perm}(N)} \frac{1}{N}\sum_{i=1}^N W_2^2(\mu^{(i)}, \mu_K^{(\sigma(i))})$$

However, it follows by the definition of optimal quantization of $\mu^{(i)}$ that, for any $1 \le j \le N$,

$$W_2^2(\mu^{(i)}, \mu_K^{(i)}) \le W_2^2(\mu^{(i)}, \mu_K^{(j)}).$$

It is then easy to see that in both cases, the optimal permutation minimizing (21) is the identity $\sigma(i) = i$ for all $1 \le i \le N$, and that we have

$$\mathcal{W}_2^2(\mathbb{P}^N, \mathbb{P}_K^N) = \frac{1}{N}\sum_{i=1}^N W_2^2(\mu^{(i)}, \mu_K^{(i)}) = \frac{1}{N}\sum_{i=1}^N \varepsilon_K(\mu^{(i)})$$

which concludes the proof.

$\square$

Similar convergence results for other downstream quantities considered in the paper (e.g., Wasserstein barycenters, dispersion, clustering performance, and LOT covariance operators) when using individual quantization.

## C   Proofs of Section 3

### C.1   Proof of Proposition 3.1

*Proof of Proposition 3.1.* Let $X \in (\mathbb{R}^d)^K$. Applying Lemma A.1, we obtain that, for any $1 \leq i \leq N$,

$$\min_{a^{(i)} \in \Sigma_K} W_2^2 \Big( \mu^{(i)}, \sum_{k=1}^K a_k^{(i)} \delta_{x_k} \Big) = W_2^2 \Big( \mu^{(i)}, \sum_{k=1}^K \mu^{(i)}(V_{x_k}) \delta_{x_k} \Big) = \sum_{k=1}^K \int_{V_{x_k}} \|x_k - y\|^2 \mathrm{d}\mu^{(i)}(y),$$

and thus we have that

$$
\begin{aligned}
\min_{a \in (\Sigma_K)^N} \frac{1}{N} \sum_{i=1}^N W_2^2 \Big( \mu^{(i)}, \sum_{k=1}^K a_k^{(i)} \delta_{x_k} \Big) &= \frac{1}{N} \sum_{i=1}^N W_2^2 \Big( \mu^{(i)}, \sum_{k=1}^K \mu^{(i)}(V_{x_k}) \delta_{x_k} \Big) \\
&= \frac{1}{N} \sum_{i=1}^N \sum_{k=1}^K \int_{V_{x_k}} \|x_k - y\|^2 \mathrm{d}\mu^{(i)}(y) \\
&= \sum_{k=1}^K \int_{V_{x_k}} \|x_k - y\|^2 \mathrm{d}\overline{\mu}(y) \\
&= W_2^2 \Big( \overline{\mu}, \sum_{k=1}^K \overline{\mu}(V_{x_k}) \delta_{x_k} \Big) \\
&= \int_{\mathbb{R}^d} \min_{1 \leq k \leq K} \{ \|x_k - y\|^2 \} \mathrm{d}\overline{\mu}(y),
\end{aligned}
$$

where we again apply Lemma A.1 to derive the last equality above and which concludes the proof.   $\square$

### C.2   Proof of Theorem 3.3

*Proof of Theorem 3.3.* For a fixed $N \geq 1$, since $\mathbb{P}^N$ and $\mathbb{P}_K^N$ are discrete uniform measures of the same size, one can actually restrict the optimization set in Equation (5) to permutations $\sigma \in \mathrm{Perm}(N)$ (Peyré & Cuturi, 2019)[Equation 2.2] in the following sense:

$$\mathcal{W}_2^2(\mathbb{P}^N, \mathbb{P}_K^N) = \min_{\sigma \in \mathrm{Perm}(N)} \frac{1}{N} \sum_{i=1}^N W_2^2(\mu^{(i)}, \mu_K^{(\sigma(i))}) \tag{21}$$

However, $\mu_K^{(i)}$ defined in Equation (13) corresponds to the discrete probability measure supported on $\bar{X}$ that best approximates $\mu^{(i)}$. In other words, $W_2^2(\mu^{(i)}, \mu_K^{(i)}) \leq W_2^2(\mu^{(i)}, \sum_{k=1}^K a_k \delta_{\bar{x}_k})$ for any weight vector $a \in \Sigma_K$. In particular, we have that, for any $1 \leq j \leq N$,

$$W_2^2(\mu^{(i)}, \mu_K^{(i)}) \leq W_2^2(\mu^{(i)}, \mu_K^{(j)}). \tag{22}$$

Using Inequality (22), it is then easy to see that the optimal permutation minimizing (21) is the identity $\sigma(i) = i$ for all $1 \leq i \leq N$, and that we have

$$\mathcal{W}_2^2(\mathbb{P}^N, \mathbb{P}_K^N) = \frac{1}{N} \sum_{i=1}^N W_2^2(\mu^{(i)}, \mu_K^{(i)})$$

We obtain from Proposition 3.1 that

$$\frac{1}{N}\sum_{i=1}^{N}W_2^2(\mu^{(i)},\mu_K^{(i)}) = W_2^2\left(\sum_{k=1}^{K}\overline{\mu}(V_{\bar{x}_k})\delta_{\bar{x}_k},\overline{\mu}\right) = \varepsilon_K(\overline{\mu}),$$

which concludes the proof. □

# D  Proofs of Section 4

## D.1  Proof of Proposition 4.1

*Proof of Proposition 4.1.* For a fixed $N$, and thanks to Theorem 3.3, we have that the sequence of probability measures $(\mathbb{P}_K^N)_{K\geq 1}$ such that $\mathbb{P}_K^N = \frac{1}{N}\sum_{i=1}^{N}\delta_{\mu_K^{(i)}} \subset \mathcal{P}(\mathcal{P}(\mathcal{X}))$ converges towards $\mathbb{P}^N$, that is $\mathcal{W}_2(\mathbb{P}_K^N,\mathbb{P}_N) \overset{K\to\infty}{\longrightarrow} 0$. Additionally, the Wasserstein barycenter of $\mathbb{P}^N$ is unique (Proposition 3.5 in Agueh & Carlier (2011)) since at least one of the probability measures $\mu^{(i)}, 1 \leq i \leq N$ is a.c. by hypothesis. Therefore, using Le Gouic & Loubes (2017)[Theorem 3], we immediately obtain that the sequence of barycenters of $(\mathbb{P}_K^N)_{K\geq 1}$ converges towards the barycenter of $\mathbb{P}^N$ in the $W_2$ distance.

□

## D.2  Proof of Proposition 4.2

*Proof of Proposition 4.2.* From the triangle inequality, one can write:

$$W_2(\mu_K^{(i)},\mu_K^{(j)}) \leq W_2(\mu_K^{(i)},\mu^{(i)}) + W_2(\mu^{(i)},\mu^{(j)}) + W_2(\mu^{(j)},\mu_K^{(j)}).$$

From Young's inequality, $2ab \leq a^2 + b^2$ and $2ab \leq \lambda a^2 + \frac{b^2}{\lambda}$ for any $\lambda > 0$ and any real numbers $a, b$ and $c$. Then it holds that :

$$(a + b + c)^2 \leq (2+\lambda)a^2 + (2+\lambda)c^2 + \left(1 + \frac{2}{\lambda}\right)b^2 \tag{23}$$

Squaring the triangle inequality and using Inequation (23) yields to:

$$W_2^2(\mu_K^{(i)},\mu_K^{(j)}) \leq (2+\lambda)W_2^2(\mu_K^{(i)},\mu^{(i)}) + \left(1 + \frac{2}{\lambda}\right)W_2^2(\mu^{(i)},\mu^{(j)}) + (2+\lambda)W_2^2(\mu^{(j)},\mu_K^{(j)}) \tag{24}$$

We now sum Inequation (24) over all $1 \leq i, j \leq N$, and divide by $N^2$:

$$\mathrm{SS}(\mu_K) = \frac{1}{N^2}\sum_{i,j=1}^{N}W_2^2(\mu_K^{(i)},\mu_K^{(j)}) \leq \frac{2}{N}(2+\lambda)\sum_{i=1}^{N}W_2^2(\mu_K^{(i)},\mu^{(i)}) + \frac{1}{N^2}\left(1 + \frac{2}{\lambda}\right)\sum_{i,j=1}^{N}W_2^2(\mu^{(i)},\mu^{(j)})$$

Hence, by Theorem 3.3, we obtain that $\mathrm{SS}(\mu_K) \leq (4+2\lambda)\varepsilon_K + \left(1 + \frac{2}{\lambda}\right)\mathrm{SS}(\mu)$, which concludes the proof. □

## D.3  Proof of Proposition 3.5

Proposition 3.5 is obtained by combining Lemmas D.1 and D.2 below.

**Lemma D.1.** *Suppose that the probability measures $(\mu^{(i)})_{i=1}^{N}$ are a.c. Then, one has for $1 \leq i, j \leq N$ and the mean-measure quantization approach in Equation (13)*

$$W_2^2(\mu_K^{(i)},\mu_K^{(j)}) \leq 3W_2^2(\mu^{(i)},\mu^{(j)}) + 6\max_{1\leq k\leq K}\mathrm{diam}(V_k),$$

*with $\mathrm{diam}(V_k) = \max_{x,y\in V_k}\|x-y\|^2$ and $(V_k)_{k=1}^{K}$ the Voronoï cells obtained from the $K$-points quantization of $\bar{\mu}$.*

*Proof of Lemma D.1.* We first recall that the dual formulation of OT between the discrete measures $\mu_K^{(i)}$ and $\mu_K^{(j)}$ (see e.g. Peyré & Cuturi (2019)) is given by

$$W_2^2(\mu_K^{(i)}, \mu_K^{(j)}) = \max_{(\alpha,\beta)\in\Phi} \sum_{k=1}^K a_k^{(i)}\alpha_k + \sum_{k=1}^K a_k^{(j)}\beta_k \tag{25}$$

where $\Phi := \{(\alpha,\beta)\in\mathbb{R}^K\times\mathbb{R}^K$ such that for all $1\le k,l\le K, \alpha_k+\beta_l \le \|\bar{x}_k - \bar{x}_l\|^2\}$. Let $\alpha^{ij}, \beta^{ij} \in \mathbb{R}^K$ be optimal Kantorovich potentials for $\mu_K^{(i)}$ and $\mu_K^{(j)}$ in Equation (25). We define the piecewise constant function $f^{ij}: \mathbb{R}^d \to \mathbb{R}$ such that $x \mapsto \alpha_k^{ij}$ when $x \in \text{int}(V_k)$, where $\text{int}(V_k)$ denotes the open interior of the Voronoï cell $V_k$. Similarly, $g^{ij}: y \mapsto \beta_k^{ij}$ when $y \in V_k$. Then, thanks to the absolute continuity of the $\mu^{(i)}$'s, one can write:

$$\begin{aligned}
W_2^2(\mu_K^{(i)}, \mu_K^{(j)}) &= \sum_{k=1}^K a_k^{(i)}\alpha_k^{ij} + \sum_{k=1}^K a_k^{(j)}\beta_k^{ij} \\
&= \sum_{k=1}^K \int_{V_k} \mathrm{d}\mu^{(i)}(x)\alpha_k^{ij} + \sum_{k=1}^K \int_{V_k} \mathrm{d}\mu^{(j)}(y)\beta_k^{ij} \\
&= \sum_{k=1}^K \int_{\text{int}(V_k)} \alpha_k^{ij}\mathrm{d}\mu^{(i)}(x) + \sum_{k=1}^K \int_{\text{int}(V_k)} \beta_k^{ij}\mathrm{d}\mu^{(j)}(y) \\
&= \int_{\mathbb{R}^d} f^{ij}(x)\mathrm{d}\mu^{(i)}(x) + \int_{\mathbb{R}^d} g^{ij}(y)\mathrm{d}\mu^{(j)}(y)
\end{aligned} \tag{26}$$

We then aim at identifying Equation (26) with a dual formulation for OT between $\mu^{(i)}$ and $\mu^{(j)}$ with respect to some cost function $c: \mathbb{R}^d \times \mathbb{R}^d \to \mathbb{R}$ to be defined later on, where

$$\begin{aligned}
\text{OT}_c(\mu^{(i)}, \mu^{(j)}) &= \min_{\pi\in\Pi(\mu^{(i)},\mu^{(j)})} \int c(x,y)\mathrm{d}\pi(x,y) \\
&= \sup_{f,g:f(x)+g(y)\le c(x,y)} \int_{\mathbb{R}^d} f(x)\mathrm{d}\mu^{(i)}(x) + \int_{\mathbb{R}^d} g(y)\mathrm{d}\mu^{(j)}(y).
\end{aligned} \tag{27}$$

If $x \in V_k$ and $y \in V_{k'}$, we obtain

$$\begin{aligned}
f^{ij}(x) + g^{ij}(y) &= \alpha_k^{ij} + \beta_{k'}^{ij} \\
&\le \|x_k - x_{k'}\|^2 \\
&\le 3\|x_k - x\|^2 + 3\|x - y\|^2 + 3\|y - x_{k'}\|^2 \\
&\le 3\text{diam}(V_k) + 3\text{diam}(V_{k'}) + 3\|x - y\|^2 \\
&\le 6\max_k \text{diam}(V_k) + 3\|x - y\|^2,
\end{aligned} \tag{28}$$

where the first inequality is due to the fact that $\alpha_k^{ij}$ and $\beta_{k'}^{ij}$ are Kantorovich potentials of $W_2(\mu_K^{(i)}, \mu_K^{(j)})$ in Equation (25). The second inequality comes from Inequality (23) where $\lambda = 1$. Defining the new cost function $c(x,y) = 6\max_k \text{diam}(V_k) + 3\|x-y\|^2$, we thus define

$$\begin{aligned}
\text{OT}_c(\mu^{(i)}, \mu^{(j)}) &= 6\max_k \text{diam}(V_k) + 3 \min_{\pi\in\Pi(\mu^{(i)},\mu^{(j)})} \int \|x-y\|^2\mathrm{d}\pi(x,y) \\
&= 6\max_k \text{diam}(V_k) + 3W_2^2(\mu^{(i)}, \mu^{(j)}).
\end{aligned}$$

Now, by Inequality (28), it follows that $f^{ij}$ and $g^{ij}$ are feasible Kantorovich potentials of $\mathrm{OT}_c$ between $\mu^{(i)}$ and $\mu^{(j)}$ in Equation (27). Hence, we finally obtain from Equation (26) that

$$
\begin{aligned}
W_2^2(\mu_K^{(i)}, \mu_K^{(j)}) &\leq \int_{\mathbb{R}^d} f^{ij}(x)\mathrm{d}\mu^{(i)}(x) + \int_{\mathbb{R}^d} g^{ij}(y)\mathrm{d}\mu^{(j)}(y) \\
&\leq \sup_{f,g:f(x)+g(y)\leq c(x,y)} \int_{\mathbb{R}^d} f(x)\mathrm{d}\mu^{(i)}(x) + \int_{\mathbb{R}^d} g(y)\mathrm{d}\mu^{(j)}(y) \\
&= 6\max_k \mathrm{diam}(V_k) + 3W_2^2(\mu^{(i)}, \mu^{(j)}),
\end{aligned}
$$

which concludes the proof. □

The following lemma provides an upper bound on the term $\max_k \mathrm{diam}(V_k)$ in Lemma D.1 in the special case where the support of $\bar{\mu}$ is included in $[0,1]^d$. This bound depends on the number of centers $K$ and the ambient dimension $d$, and holds true for either discrete or continuous support.

**Lemma D.2.** *Suppose that the (discrete or continuous) support of the mean measure $\bar{\mu}$ is included in $[0,1]^d$ and let $(V_k)_{k=1}^K$ be the Voronoï cells of the quantization of $\bar{\mu}$. Then,*

$$
\max_{1\leq k\leq K} \mathrm{diam}(V_k) \leq \frac{d}{\lfloor \sqrt[d]{K} \rfloor^2}.
$$

*Proof of Lemma D.2.* Suppose that the support of the mean measure $\bar{\mu}$ is included in $[0,1]^d$. Then, we first have that $\max_{1\leq k\leq K} \mathrm{diam}(V_k) \leq \max_{1\leq j\leq \lfloor \sqrt[d]{K}\rfloor^d} \mathrm{diam}(V_j)$. Indeed, as $\lfloor \sqrt[d]{K}\rfloor^d \leq K$, this is simply reducing the number of quantization points, and therefore increasing the maximum diameter of the cells. Now, denoting $K' = \lfloor \sqrt[d]{K}\rfloor^d$ the $d$-th power of the integer $\lfloor \sqrt[d]{K}\rfloor$, one can grid the support space $[0,1]^d$ with $K'$ points $\{x_1, \ldots, x_{K'}\}$ set as $\left\{ \left( \frac{a_1^{(i)}}{\lfloor \sqrt[d]{K}\rfloor}, \cdots, \frac{a_d^{(i)}}{\lfloor \sqrt[d]{K}\rfloor} \right) \mid a_k^{(i)} \in \{1, \cdots, d\} \right\}$. With these centers, all Voronoï cells have the same diameter, which is:

$$
\forall 1\leq k\leq K, \ \mathrm{diam}(V_k) = \left\| \left( \frac{1}{\lfloor \sqrt[d]{K}\rfloor}, \cdots, \frac{1}{\lfloor \sqrt[d]{K}\rfloor} \right) \right\|^2 = \sum_{i=1}^d \left( \frac{1}{\lfloor \sqrt[d]{K}\rfloor} \right)^2 = \frac{d}{\lfloor \sqrt[d]{K}\rfloor^2}.
$$

This finally gives us:

$$
\max_{1\leq k\leq K} \mathrm{diam}(V_k) \leq \frac{d}{\lfloor \sqrt[d]{K}\rfloor^2}.
$$

□

## D.4 Proof of Proposition 4.3

*Proof of Proposition 4.3.* For the result regarding the within-class variance of the clusters, we have, using the triangle inequality and (23) with $\lambda = 1$,

$$
W_2^2(\mu_K^{(i)}, \mu_K^{(j)}) \leq 3\left( W_2^2(\mu_K^{(i)}, \mu^{(i)}) + W_2^2(\mu^{(i)}, \mu^{(j)}) + W_2^2(\mu^{(j)}, \mu_K^{(j)}) \right)
$$

Summing over the indices of $I_l$ and dividing by $N_l^2$ yields:

$$
\begin{aligned}
\mathrm{WCSS}(l, \mu_K) &\leq \frac{3}{N_l^2} \sum_{i,j\in I_l} W_2^2(\mu_K^{(i)}, \mu^{(i)}) + \frac{3}{N_l^2} \sum_{i,j\in I_l} W_2^2(\mu^{(i)}, \mu^{(j)}) + \frac{3}{N_l^2} \sum_{i,j\in I_l} W_2^2(\mu^{(j)}, \mu_K^{(j)}) \\
&\leq \frac{6}{N_l} \sum_{i\in I_l} W_2^2(\mu_K^{(i)}, \mu^{(i)}) + 3\mathrm{WCSS}(l, \mu) \\
&\leq \frac{6}{N_l} \sum_{1\leq i\leq N} W_2^2(\mu_K^{(i)}, \mu^{(i)}) + 3\mathrm{WCSS}(l, \mu) = \frac{6N}{N_l}\varepsilon_K + 3\mathrm{WCSS}(l, \mu),
\end{aligned}
$$

where the last equality follows from Theorem 3.3, which concludes the first item (14) of the proposition. For the second statement on the between-class variance, we rewrite the triangle inequality:

$$3\big(W_2^2(\mu_K^{(i)}, \mu^{(i)}) + W_2^2(\mu_K^{(i)}, \mu_K^{(j)}) + W_2^2(\mu^{(j)}, \mu_K^{(j)})\big) \geq W_2^2(\mu^{(i)}, \mu^{(j)})$$

$$\Leftrightarrow W_2^2(\mu_K^{(i)}, \mu_K^{(j)}) \geq \frac{1}{3} W_2^2(\mu^{(i)}, \mu^{(j)}) - W_2^2(\mu_K^{(i)}, \mu^{(i)}) - W_2^2(\mu^{(j)}, \mu_K^{(j)})$$

Summing over the indices of $I_{l_1}$ and $I_{l_2}$ and dividing by $N_{l_1} N_{l_2}$ gives:

$$
\begin{aligned}
\text{BCSS}(l_1, l_2, \mu_K) &\geq \frac{1}{3} \frac{1}{N_{l_1} N_{l_2}} \sum_{\substack{i_1 \in I_{l_1} \\ i_2 \in I_{l_2}}} W_2^2(\mu^{(i_1)}, \mu^{(i_2)}) \\
&\quad - \frac{1}{N_{l_1} N_{l_2}} \sum_{\substack{i_1 \in I_{l_1} \\ i_2 \in I_{l_2}}} W_2^2(\mu_K^{(i_1)}, \mu^{(i_1)}) - \frac{1}{N_{l_1} N_{l_2}} \sum_{\substack{i_1 \in I_{l_1} \\ i_2 \in I_{l_2}}} W_2^2(\mu^{(i_2)}, \mu_K^{(i_2)}) \\
&\geq \frac{1}{3} \text{BCSS}(l_1, l_2, \mu) - \frac{1}{N_{l_1} N_{l_2}} \sum_{1 \leq i_1, i_2 \leq N} W_2^2(\mu_K^{(i_1)}, \mu^{(i_1)}) \\
&\quad - \frac{1}{N_{l_1} N_{l_2}} \sum_{1 \leq i_1, i_2 \leq N} W_2^2(\mu_K^{(i_2)}, \mu^{(i_2)}) \\
&= \frac{1}{3} \text{BCSS}(l_1, l_2, \mu) - \frac{1}{N_{l_1}} \sum_{1 \leq i_1 \leq N} W_2^2(\mu_K^{(i_1)}, \mu^{(i_1)}) \\
&\quad - \frac{1}{N_{l_2}} \sum_{1 \leq i_2 \leq N} W_2^2(\mu_K^{(i_2)}, \mu^{(i_2)}) \\
&= \frac{1}{3} \text{BCSS}(l_1, l_2, \mu) - \Big(\frac{1}{N_{l_1}} + \frac{1}{N_{l_2}}\Big) \sum_{1 \leq i \leq N} W_2^2(\mu_K^{(i)}, \mu^{(i)}) \\
&= \frac{1}{3} \text{BCSS}(l_1, l_2, \mu) - \Big(\frac{N}{N_{l_1}} + \frac{N}{N_{l_2}}\Big) \varepsilon_K,
\end{aligned}
$$

where the last equality follows from Theorem 3.3, which concludes the proof. $\qquad\square$

### D.5 Proof of Proposition 4.4

We start by proving Lemma D.3 below, which holds for an arbitrary Hilbert space embedding that is $\alpha$-Holder continuous with regards to the 2-Wasserstein distance. Let $\mathcal{H}$ be a Hilbert space with corresponding norm $\|\cdot\|_{\mathcal{H}}$. Let $\Phi : \mathcal{P}(\mathcal{X}) \to \mathcal{H}$ be a Hilbert space embedding of probability measures. We define the empirical covariance operators of the embedded original measures and the embedded quantized measures as

$$\Sigma^N = \frac{1}{N} \sum_{i=1}^{N} \Phi(\mu^{(i)}) \otimes \Phi(\mu^{(i)}) \qquad \Sigma_K^N = \frac{1}{N} \sum_{i=1}^{N} \Phi(\mu_K^{(i)}) \otimes \Phi(\mu_K^{(i)}).$$

**Lemma D.3.** *Let $\Phi : \mathcal{P}(\mathcal{X}) \to \mathcal{H}$ be a Hilbert space embedding such that $\|\Phi(\mu)\|_{\mathcal{H}} \leq R$ for any $\mu \in \mathcal{P}(\mathcal{X})$. For $0 < \alpha \leq 1$, assume $\Phi$ is $\alpha$-Holder continuous with regards to the 2-Wasserstein distance, that is, there exists a constant $C > 0$ such that*

$$\forall \mu, \nu \in \mathcal{P}(\mathcal{X}), \qquad \|\Phi(\mu) - \Phi(\nu)\|_{\mathcal{H}} \leq C W_2^\alpha(\mu, \nu).$$

*Then,*

$$\|\Sigma^N - \Sigma_K^N\|_{\text{HS}} \leq 2RC \, \varepsilon_K^{\alpha/2}(\overline{\mu}).$$

*Proof of Lemma D.3.* We can write

$$\|\Sigma^N - \Sigma_K^N\|_{\mathrm{HS}} = \frac{1}{N}\Big\|\sum_{i=1}^N \Phi(\mu^{(i)}) \otimes \Phi(\mu^{(i)}) - \Phi(\mu_K^{(i)}) \otimes \Phi(\mu_K^{(i)})\Big\|_{\mathrm{HS}}$$

$$\leq \frac{1}{N}\sum_{i=1}^N \Big\|\Phi(\mu^{(i)}) \otimes \Phi(\mu^{(i)}) - \Phi(\mu_K^{(i)}) \otimes \Phi(\mu_K^{(i)})\Big\|_{\mathrm{HS}}$$

One has that

$$\Big\|\Phi(\mu^{(i)}) \otimes \Phi(\mu^{(i)}) - \Phi(\mu_K^{(i)}) \otimes \Phi(\mu_K^{(i)})\Big\|_{\mathrm{HS}} = \Big\|\big(\Phi(\mu^{(i)}) - \Phi(\mu_K^{(i)})\big) \otimes \Phi(\mu^{(i)}) + \Phi(\mu_K^{(i)}) \otimes \Phi(\mu^{(i)}) - \Phi(\mu_K^{(i)}) \otimes \Phi(\mu_K^{(i)})\Big\|_{\mathrm{HS}}$$

$$= \Big\|\big(\Phi(\mu^{(i)}) - \Phi(\mu_K^{(i)})\big) \otimes \Phi(\mu^{(i)}) + \Phi(\mu_K^{(i)}) \otimes \big(\Phi(\mu^{(i)}) - \Phi(\mu_K^{(i)})\big)\Big\|_{\mathrm{HS}}$$

$$\leq \Big\|\big(\Phi(\mu^{(i)}) - \Phi(\mu_K^{(i)})\big) \otimes \Phi(\mu^{(i)})\Big\|_{\mathrm{HS}} + \Big\|\Phi(\mu_K^{(i)}) \otimes \big(\Phi(\mu^{(i)}) - \Phi(\mu_K^{(i)})\big)\Big\|_{\mathrm{HS}}$$

Using that $\|u \otimes v\|_{\mathrm{HS}} = \|u\|_{\mathcal{H}}\|v\|_{\mathcal{H}}$ for any $u, v \in \mathcal{H}$, we obtain

$$\Big\|\Phi(\mu^{(i)}) \otimes \Phi(\mu^{(i)}) - \Phi(\mu_K^{(i)}) \otimes \Phi(\mu_K^{(i)})\Big\|_{\mathrm{HS}} \leq \|\Phi(\mu^{(i)}) - \Phi(\mu_K^{(i)})\|_{\mathcal{H}}\|\Phi(\mu^{(i)})\|_{\mathcal{H}} + \|\Phi(\mu_K^{(i)})\|_{\mathcal{H}}\|\Phi(\mu^{(i)}) - \Phi(\mu_K^{(i)})\|_{\mathcal{H}}$$

$$= 2R\|\Phi(\mu^{(i)}) - \Phi(\mu_K^{(i)})\|_{\mathcal{H}},$$

where $R$ is an upperbound on $\|\Phi(\mu)\|_{\mathcal{H}}$ for any $\mu \in \mathcal{P}(\mathcal{X})$. Therefore, we have that

$$\|\Sigma^N - \Sigma_K^N\|_{\mathrm{HS}} \leq \frac{2R}{N}\sum_{i=1}^N \|\Phi(\mu^{(i)}) - \Phi(\mu_K^{(i)})\|_{\mathcal{H}}$$

Using the $\alpha$-Holder continuity of $\Phi$ gives

$$\|\Sigma^N - \Sigma_K^N\|_{\mathrm{HS}} \leq \frac{2R}{N}\sum_{i=1}^N C W_2^\alpha(\mu^{(i)}, \mu_K^{(i)})$$

$$= \frac{2RC}{N}\sum_{i=1}^N \Big(W_2^2(\mu^{(i)}, \mu_K^{(i)})\Big)^{\alpha/2}$$

As $\alpha/2 \leq 1$, we can use Jensen's inequality to obtain

$$\|\Sigma^N - \Sigma_K^N\|_{\mathrm{HS}} \leq 2RC\Big(\frac{1}{N}\sum_{i=1}^N W_2^2(\mu^{(i)}, \mu_K^{(i)})\Big)^{\alpha/2}$$

$$= 2RC\varepsilon_K^{\alpha/2}(\overline{\mu}),$$

which concludes the proof. □

The LOT embedding is $\Phi^{\text{LOT}} : \mathcal{P}(\mathcal{X}) \mapsto T_\rho^\mu - \text{Id} \in L^2(\rho)$ for an a.c. reference measure $\rho$. Note that for any measure $\mu \in \mathcal{P}(\mathcal{X})$, $\|\Phi^{\text{LOT}}(\mu)\|_{L^2(\rho)}^2 = \int |T_\rho^\mu(x) - x|^2 \mathrm{d}\rho(x) \leq \int \text{diam}(\mathcal{X})^2 \mathrm{d}\rho(x) = \text{diam}(\mathcal{X})^2$. The LOT embedding is therefore bounded with $R = \text{diam}(\mathcal{X})$. We now use a result due to Ambrosio and reported in Gigli (2011) and Delalande & Mérigot (2023)[Theorem (Ambrosio)], which states that when $\rho$ is a probability density over a compact set $\mathcal{X}$, $\mu$ and $\nu$ are probability measures on a $\mathcal{X}$ and $T_\rho^\mu$ is $L$-Lipschitz (by hypothesis), then:

$$\|T_\rho^\mu - T_\rho^\nu\|_{L^2(\rho)} \leq 2\sqrt{\text{diam}(\mathcal{X})L} W_1(\mu,\nu)^{1/2}. \tag{29}$$

Using that $W_1(\mu,\nu) \leq W_2(\mu,\nu)$ for any $\mu, \nu \in \mathcal{P}(\mathcal{X})$, we deduce from (29) that the LOT embedding is $\alpha$-Holder continuous with $\alpha = 1/2$ and $C = 2\sqrt{\text{diam}(\mathcal{X})L}$. Therefore, we can apply Lemma D.3 to obtain

$$\|\Sigma^N - \Sigma_K^N\|_{\text{HS}} \leq 4\text{diam}(\mathcal{X})^{3/2} L^{1/2} \, \varepsilon_K^{1/4}(\overline{\mu}).$$

### D.6 Extension of Proposition 4.4 to the kernel mean embedding

Given a positive definite kernel function $k : \mathcal{X} \times \mathcal{X} \to \mathbb{R}$ and associated RKHS $\mathcal{H}$, the kernel mean embedding (KME) is defined as:

$$\Phi^{\text{KME}} : \mathcal{P}(\mathcal{X}) \to \mathcal{H}$$

$$\mu \to \int_{\mathcal{X}} k(x,\cdot)\mathrm{d}\mu(x).$$

We derive a similar result to Proposition 4.4 for the KME.

**Proposition D.4.** *Let $\Sigma^N$ and $\Sigma_K^N$ be the covariance operators associated with the KME of the orginal measures and the quantized measures respectively. Assume that there exists a constant $c > 0$ such that*

$$\forall x, y \in \mathcal{X}, \qquad k(x,x) + k(y,y) - 2k(x,y) \leq c^2 \|x-y\|^2.$$

*Assume also that the kernel $k$ is bounded by a constant $M_k < \infty$. Then,*

$$\|\Sigma^N - \Sigma_K^N\|_{\text{HS}} \leq 2Rc\varepsilon_K^{1/2}(\overline{\mu}),$$

*Proof of Proposition D.4.* For any $\mu \in \mathcal{P}(\mathcal{X})$, we have that:

$$\|\Phi^{\text{KME}}(\mu)\|_{\mathcal{H}}^2 = \langle \int_{\mathcal{X}} k(x,\cdot)\mathrm{d}\mu(x), \int_{\mathcal{X}} k(y,\cdot)\mathrm{d}\mu(y) \rangle_{\mathcal{H}} = \int_{\mathcal{X} \times \mathcal{X}} k(x,y)\mathrm{d}\mu(x)\mathrm{d}\mu(y) \leq M_k.$$

The KME is therefore bounded. Furthermore, we know that $(\mathcal{X}, \|\cdot\|)$ is a complete separable metric space, $k$ is positive definite and verifies $\forall x, y \in \mathcal{X}, k(x,x) + k(y,y) - 2k(x,y) \leq c^2\|x-y\|^2$ for a constant $c > 0$, and $\mu^{(i)}$ and $\mu_K^{(i)}$ have finite 2-moments as they are supported on a compact set $\mathcal{X} \subset \mathbb{R}^d$. We can then use Proposition 2 of Vayer & Gribonval (2023), which proves that under these assumptions,

$$\|\Phi^{\text{KME}}(\mu^{(i)}) - \Phi^{\text{KME}}(\mu_K^{(i)})\|_{\mathcal{H}} \leq cW_2(\mu^{(i)}, \mu_K^{(i)}).$$

The embedding $\Phi^{\text{KME}}$ is therefore 1-Holder continuous with respect to the 2-Wasserstein distance. Applying Lemma D.3 with $\alpha = 1$ and $C = c$ yields,

$$\|\Sigma^N - \Sigma_K^N\|_{\text{HS}} \leq 2Rc\varepsilon_K^{1/2}(\overline{\mu}),$$

which concludes the proof. □

# E  Convergence of the Gram matrix

We denote $G^N$ and $G_K^N$ the Gram matrices associated to the LOT embeddings of the original measures and the quantized measures respectively, that is

$$\forall 1 \leq i,j \leq N, \qquad G_{ij}^N = \langle \Phi^{\mathrm{LOT}}(\mu^{(i)}), \Phi^{\mathrm{LOT}}(\mu^{(j)}) \rangle_{L^2(\rho)}, \quad (G_K^N)_{ij} = \langle \Phi^{\mathrm{LOT}}(\mu_K^{(i)}), \Phi^{\mathrm{LOT}}(\mu_K^{(j)}) \rangle_{L^2(\rho)}.$$

We denote $\|\cdot\|_F$ the Frobenius norm on $\mathbb{R}^{N\times N}$.

**Proposition E.1** (Convergence of the Gram matrix). *Let $G^N$ and $G_K^N$ be the Gram matrices associated to the LOT embeddings of the original measures and the quantized measures respectively. Assume that the optimal transport maps $T_\rho^{\mu^{(i)}}$ are $L$-Lipschitz for any $1 \leq i \leq N$. Then,*

$$\|G^N - G_K^N\|_F \leq 4N\,\mathrm{diam}(\mathcal{X})^{3/2} L^{1/2}\, \varepsilon_K^{1/4}(\overline{\mu}).$$

As a consequence of Proposition E.1, we have that any ML tasks relying on the Gram matrix $G_K^N$ is consistent (as $K \to \infty$) with the one relying on $G^N$.

*Proof of Proposition E.1.* We start by proving Lemma E.2 below, which holds for an arbitrary Hilbert space embedding that is $\alpha$-Holder continuous with regards to the 2-Wasserstein distance. Let $\mathcal{H}$ be a Hilbert space with corresponding inner-product $\langle \cdot, \cdot \rangle_{\mathcal{H}}$ and norm $\|\cdot\|_{\mathcal{H}}$. Let $\Phi : \mathcal{P}(\mathcal{X}) \to \mathcal{H}$ be a Hilbert space embedding of probability measures. We define the Gram matrices of the embedded original measures and embedded quantized measures as

$$\forall 1 \leq i,j \leq N, \qquad G_{ij}^N = \langle \Phi(\mu^{(i)}), \Phi(\mu^{(j)}) \rangle_{\mathcal{H}}, \quad (G_K^N)_{ij} = \langle \Phi(\mu_K^{(i)}), \Phi(\mu_K^{(j)}) \rangle_{\mathcal{H}}.$$

**Lemma E.2** (Convergence of the Gram matrix). *Let $\Phi : \mathcal{P}(\mathcal{X}) \to \mathcal{H}$ be a Hilbert space embedding such that $\|\Phi(\mu)\| \leq R$ for any $\mu \in \mathcal{P}(\mathcal{X})$. Assume $\Phi$ is $\alpha$-Holder continuous with respect to the 2-Wasserstein distance for $0 < \alpha \leq 1$. Then,*

$$\|G^N - G_K^N\|_F \leq 2RCN\varepsilon_K^{\alpha/2}(\overline{\mu}).$$

*Proof of Lemma E.2.* We can write

$$\|G^N - G_K^N\|_F^2 = \sum_{i,j=1}^N \left( \langle \Phi(\mu^{(i)}), \Phi(\mu^{(j)}) \rangle_{\mathcal{H}} - \langle \Phi(\mu_K^{(i)}), \Phi(\mu_K^{(j)}) \rangle_{\mathcal{H}} \right)^2$$

One has that

$$
\begin{aligned}
\left( \langle \Phi(\mu^{(i)}), \Phi(\mu^{(j)}) \rangle_{\mathcal{H}} - \langle \Phi(\mu_K^{(i)}), \Phi(\mu_K^{(j)}) \rangle_{\mathcal{H}} \right)^2 &= \left( \langle \Phi(\mu^{(i)}) - \Phi(\mu_K^{(i)}), \Phi(\mu^{(j)}) \rangle_{\mathcal{H}} + \langle \Phi(\mu_K^{(i)}), \Phi(\mu^{(j)}) \rangle_{\mathcal{H}} - \langle \Phi(\mu_K^{(i)}), \Phi(\mu_K^{(j)}) \rangle_{\mathcal{H}} \right)^2 \\
&= \left( \langle \Phi(\mu^{(i)}) - \Phi(\mu_K^{(i)}), \Phi(\mu^{(j)}) \rangle_{\mathcal{H}} + \langle \Phi(\mu_K^{(i)}), \Phi(\mu^{(j)}) - \Phi(\mu_K^{(j)}) \rangle_{\mathcal{H}} \right)^2 \\
&\leq 2\langle \Phi(\mu^{(i)}) - \Phi(\mu_K^{(i)}), \Phi(\mu^{(j)}) \rangle_{\mathcal{H}}^2 + 2\langle \Phi(\mu_K^{(i)}), \Phi(\mu^{(j)}) - \Phi(\mu_K^{(j)}) \rangle_{\mathcal{H}}^2 \\
&= 2\|\Phi(\mu^{(i)}) - \Phi(\mu_K^{(i)})\|_{\mathcal{H}}^2 \|\Phi(\mu^{(j)})\|_{\mathcal{H}}^2 + 2\|\Phi(\mu_K^{(i)})\|_{\mathcal{H}}^2 \|\Phi(\mu^{(j)}) - \Phi(\mu_K^{(j)})\|_{\mathcal{H}}^2 \\
&\leq 2R^2 \|\Phi(\mu^{(i)}) - \Phi(\mu_K^{(i)})\|_{\mathcal{H}}^2 + 2R^2 \|\Phi(\mu^{(j)}) - \Phi(\mu_K^{(j)})\|_{\mathcal{H}}^2.
\end{aligned}
$$

Therefore, we have that

$$\|G^N - G_K^N\|_F^2 \leq \sum_{i,j=1}^{N} 2R^2 \|\Phi(\mu^{(i)}) - \Phi(\mu_K^{(i)})\|_{\mathcal{H}}^2 + \sum_{i,j=1}^{N} 2R^2 \|\Phi(\mu^{(j)}) - \Phi(\mu_K^{(j)})\|_{\mathcal{H}}^2$$

$$= 2R^2 N \sum_{i=1}^{N} \|\Phi(\mu^{(i)}) - \Phi(\mu_K^{(i)})\|_{\mathcal{H}}^2 + 2R^2 N \sum_{j=1}^{N} \|\Phi(\mu^{(j)}) - \Phi(\mu_K^{(j)})\|_{\mathcal{H}}^2$$

$$= 4R^2 N \sum_{i=1}^{N} \|\Phi(\mu^{(i)}) - \Phi(\mu_K^{(i)})\|_{\mathcal{H}}^2$$

Now, if the distance between the embeddings is $\alpha$-Holder continuous with respect to the 2-Wasserstein distance, then there exists a constant $C > 0$ such that for any $\mu, \nu \in \mathcal{P}(\mathcal{X})$,

$$\|\Phi(\mu) - \Phi(\nu)\|_{\mathcal{H}} \leq C W_2^\alpha(\mu, \nu).$$

Then, we have that

$$\|G^N - G_K^N\|_F^2 \leq 4R^2 N \sum_{i=1}^{N} C^2 W_2^{2\alpha}(\mu^{(i)}, \mu_K^{(i)})$$

$$= 4R^2 C^2 N \sum_{i=1}^{N} \left( W_2^2(\mu^{(i)}, \mu_K^{(i)}) \right)^\alpha$$

As $\alpha \leq 1$, we can use Jensen's inequality to obtain

$$\|G^N - G_K^N\|_F^2 \leq 4R^2 C^2 N \cdot N \left( \frac{1}{N} \sum_{i=1}^{N} W_2^2(\mu^{(i)}, \mu_K^{(i)}) \right)^\alpha$$

$$= 4R^2 C^2 N^2 \varepsilon_K^\alpha(\overline{\mu}).$$

Taking the square root yields

$$\|G^N - G_K^N\|_F \leq 2RCN \varepsilon_K^{\alpha/2}(\overline{\mu}),$$

which concludes the proof.

$\square$

For the LOT embedding, we can use the same constants as in the previous section: it is bounded with $R = \text{diam}(\mathcal{X})$ and it is $\alpha$-Holder continuous with $\alpha = 1/2$ and $C = 2\sqrt{\text{diam}(\mathcal{X})L}$. Therefore, applying Lemma E.2 to the LOT embedding yields

$$\|G^N - G_K^N\|_F \leq 4N \, \text{diam}(\mathcal{X})^{3/2} L^{1/2} \, \varepsilon_K^{1/4}(\overline{\mu}).$$

$\square$

