# OpenReview forum: "Scalable Learning from Probability Measures with Mean Measure Quantization"
_TMLR — Decision pending for TMLR_

### Review · Reviewer_bbMz · 2026-04-16

**Summary Of Contributions:**

The paper tackles the computational bottleneck of applying Optimal Transport to large-scale datasets where each observation is a probability measure with a massive support. Standard Optimal Transport methods scale poorly with the number of points per measure. To address this, the authors introduce Mean Measure Quantization. Instead of subsampling randomly or quantizing each measure independently, Mean Measure Quantization computes a shared set of K discrete support atoms by quantizing the pooled empirical mean of all input measures. Each individual measure is then approximated by assigning its mass to these shared atoms via Voronoï cell counting.

## Key Contributions
1.The authors demonstrate that shared-support quantization is mathematically equivalent to the optimal quantization of the mean measure. They establish rigorous consistency results, proving that the Wasserstein distance between the distributions of quantized measures and the original measures is bounded by the quantization error of the mean measure.
2.The paper theoretically guarantees that substituting original measures with their Mean Measure Quantization approximations preserves key statistics of Optimal Transport. Specifically, they provide convergence bounds for Wasserstein barycenters, statistical dispersion, clustering metrics, and the covariance operators/Gram matrices of Linearized Optimal Transport embeddings.
3.To avoid the massive cost of pooling millions of points, the authors propose a subsampled variant of Mean Measure Quantization. By computing the shared atoms on a small random subset of the pooled data, the computational complexity is drastically reduced while downstream performance is maintained.

## Shortcomings
1. Both the theoretical quantization error bounds and the practical k-means clustering step suffer in high-dimensional spaces. The method may require an impractically large number of centers K to maintain low distortion when the ambient dimension d is very high.
2. While shared supports align clusters globally, standard k-means heavily favors dense regions. Consequently, rare but highly discriminative subpopulations present only in a few outlier measures might be absorbed into larger Voronoï cells, especially when using the subsampled Mean Measure Quantization variant.

**Additional Comments:**

No additional comments.

**Audience:**

Yes

**Audience Explanation:**

Yes, this is very relative to researchers interested in optimal transport/ data science.

**Broader Impact Concerns:**

None.

**Claims And Evidence:**

Yes

**Claims Explanation:**

Overall, the claims made in the submission are well-supported by both theoretical derivations and empirical results. The methodology is sound.

**Requested Changes:**

1. To combat the curse of dimensionality, future research could project the measures onto a lower-dimensional latent space before applying Mean Measure Quantization, or extend the framework to utilize Sliced-Wasserstein distances.
2. The quantization step could incorporate density-based or weighted k-means algorithms that penalize the collapse of low-density, isolated data points ,in order to better preserve rare subpopulations.

---

> ### Author Response · Authors · 2026-06-16
>
> Thank you for your positive feedback and comments.
>
> **About the curse of dimensionality**
> > As pointed out by the reviewer, one limitation of quantization is its dependence to the curse of dimensionality, which affects the quantization error and the practical k-means computation. We agree that the projection-based approaches, taking inspiration from sliced optimal transport, could provide an interesting direction to tackle these issues. We have added a comment in this regard in the conclusion.
>
> **About the use of standard k-means**
> > We agree that, since our method is based on quantization of the mean-measure, extremely small subpopulations might go undetected when their contribution is negligible. We note, however, that the rare-population experiment in Section 5.1.2 shows that MMQ can still capture rare populations when they are sufficiently represented in at least some of the measures, thanks to the shared-support strategy.
>
> >Designing shared-support constructions that explicitly preserve rare but highly discriminative subpopulations would be an interesting direction for future work. Such approaches would likely require an alternative quantization objective.

---

### Review · Reviewer_Tgpg · 2026-04-27

**Summary Of Contributions:**

The authors propose a shared-support quantization method, termed Mean-Measure Quantization (MMQ), to scale up Optimal Transport (OT) based learning algorithms for collections of large-support probability measures. The core theoretical contribution establishes that finding a global shared support across multiple measures is mathematically equivalent to performing optimal quantization on their mean measure (Proposition 3.1). Building upon this, the authors prove consistency and convergence for several downstream OT tasks, such as Wasserstein barycenters, statistical dispersion, and linearized OT embeddings. Empirically, the paper demonstrates that a subsampled version of MMQ achieves classification accuracy comparable to Individual Quantization (IQ) while yielding significant computational speedups on flow cytometry and satellite imagery datasets.


### Strengths

* **Soundness of the Core Equivalence:** The equivalence established in Proposition 3.1 is conceptually elegant and theoretically sound. It successfully bridges shared-support optimization with standard mean-measure quantization.

* **High Interpretability and Practical Value:** Forcing a shared support across measures brings a major practical advantage: distributions become directly comparable via their weight vectors. The synthetic experiment involving rare populations (Figure 2) effectively highlights how this shared base helps in tracking vanishing subpopulations, an advantage not present in independent quantization.

* **Convincing Empirical Speedup:** The introduction of "Subsampled MMQ" is a highly practical engineering solution. The experiments on real-world datasets clearly validate that one can heavily subsample the mean measure without sacrificing downstream classification accuracy, thoroughly addressing the "scalability" claim in the title.

* **Clarity:** The paper is well-structured, mathematical notations are generally clean, and the narrative flows logically from theoretical equivalence to downstream applications and finally to empirical validation.

### Weaknesses

* **Gap Between Theory and the "Subsampled" Practice:** While the paper's practical speedup relies heavily on Subsampled MMQ, the theoretical consistency (Theorem 3.3 and Section 4) is built entirely on the full empirical mean measure. The paper lacks a finite-sample error analysis bounding the gap between the subsampled mean measure and the true mean measure.

* **Strong Regularity Assumptions:** The theoretical bound for the LOT embedding covariance operator (Proposition 4.4) relies on the strong assumption that the Monge maps are L-Lipschitz. In general OT theory, this requires highly restrictive conditions (e.g., Caffarelli's regularity theory bounds, requiring convex support and bounded densities).

* **Curse of Dimensionality:** As acknowledged by the authors in the conclusion, the quantization error $\epsilon_{K}(\bar{\mu})$  heavily depends on the ambient dimension d, meaning the derived bounds become extremely loose in high-dimensional settings.

**Audience:**

Yes

**Audience Explanation:**

It would be interested to researchers dedicated to the theory and application of scalable machine learning and Optimal Transport.

**Broader Impact Concerns:**

Not applicable.

**Claims And Evidence:**

Yes

**Claims Explanation:**

Yes, the claims are generally well-supported by accurate, convincing, and clear evidence.

**Strong Theoretical Evidence:** The core mathematical equivalence between shared-support optimization and mean-measure quantization is rigorously proven in Appendix C.1. The consistency guarantees derived from this for downstream tasks (such as Wasserstein barycenters and LOT embeddings) logically align perfectly with existing OT theoretical frameworks.

**Convincing Empirical Evidence:** The authors provide detailed experimental support. The synthetic datasets clearly and intuitively demonstrate that the method offers superior interpretability when tracking "rare populations." Furthermore, on the real-world Airbus satellite imagery and flow cytometry datasets, the comparison of classification accuracy and runtime explicitly confirms that Subsampled MMQ drastically reduces computation time with almost no loss in accuracy. This provides the strongest empirical support for the paper's claim of "scalability."

**Requested Changes:**

1. **Explicit Discussion on Subsampling Theory:** Please add a brief discussion acknowledging the theoretical gap regarding Subsampled MMQ. Explicitly state that the current consistency proofs apply to the full mean measure and that subsampling introduces an additional approximation error not covered in Section 3 and 4.
2. **Clarification on Lipschitz Assumptions:** Add a short remark after Proposition 4.4 discussing the restrictiveness of the L-Lipschitz Monge map assumption in practical scenarios.
3. **Fix Typo in Appendix C.1:** In the first line of the proof for Proposition 3.1, there is a missing superscript `(i)`. The equation is written as $W_2^2(\mu, \sum_{k=1}^K a_k^{(i)}\delta_{x_k})$ but should be $W_2^2(\mu^{(i)}, \sum_{k=1}^K a_k^{(i)}\delta_{x_k})$. This typo also carries over to the subsequent summation line. Please correct this.

---

> ### Author Response · Authors · 2026-06-16
>
> Thank you for your time and feedback. Below we address each concern.
>
> **On the gap between theory and the subsampling practice**
> > We thank the reviewer for this insightful comment. We agree that the theoretical analysis conducted in Sections 3 and 4 only applies to MMQ constructed from the full mean measure. While the numerical experiments show that subsampled MMQ can reduce computational time without noticeable loss in performance, the additional approximation error introduced by the subsampling step is not covered by our current analysis. We have added a remark in Section 5.2 explicitly acknowledging this gap.
>
> **On the regularity assumptions on the Monge maps**
> > We thank the reviewer for this remark. We agree that the assumption that the Monge maps $T_{\rho}^{\mu^{(i)}}$ are $L$-Lipschitz is restrictive. Such regularity typically requires strong assumptions on the measures, for instance convex supports and densities bounded from above and below. In particular, this assumption is not satisfied for discrete measures. Alternative stability results for optimal transport maps are available under different assumptions [1]. This can lead to variant of Proposition 4.4 under weaker assumptions, and with different exponents. We have added a short remark after Proposition 4.4 to clarify this point.
>
> **About the curse of dimensionality**
> > We agree with the reviewer that the quantization error $\varepsilon_K(\overline{\mu})$ suffers from the curse of dimensionality, as the classical quantization rates scale exponentially with the ambient dimension. This limitation is shared by many OT-based statistical problems. We have already highlighted this point in the conclusion  and now emphasize it more clearly.
>
> **Typos in Appendix C.1**
> > We thank the reviewer for pointing this out. We have corrected this issue in the revision.
>
> [1] Merigot, Quentin. "Sharp stability of Brenier maps via quantitative regularity of potentials." (2026).

---

> > ### Comment · Reviewer_Tgpg · 2026-06-19
> >
> > Thank you for the response. My comments have been adequately addressed.

---

### Review · Reviewer_yA2a · 2026-06-07

**Summary Of Contributions:**

This paper considers the problem of approximating probability distributions via quantization, with the goal of making downstream optimal transport applications more feasible. The basic setup is this: you have $N$ distributions (in this work, they are discrete distributions corresponding to a large number of data points) and want to approximate each of them with a discrete distribution over $K$ points. This approximate distribution is represented by $K$ vectors for the support and $K$ weights over that support. The problem considered in this work is what happens when we only allow ourselves to use the same $K$ points for all $N$ distributions (but allow the weights to vary).

The central theoretical result shows how this problem relates to quantizing the average measure over all data points. Then the authors give some theoretical results about eventual consistency of these approximations to downstream tasks and experiments on real and synthetic data showing the method is practical.

**Additional Comments:**

An AI review tool (like Refine, or sufficient prompting with ChatGPT) would likely have flagged the $K\to\infty$ issue. They are very good at catching when assumptions don't quite line up.

I was confused by the discussion around linearized OT (which I have never seen before) and think that section could be improved. Three concrete questions:
1. What is Brenier's theorem? You should either include it or point the reader to a specific theorem. You cite Brenier's paper, but I don't know which theorem you mean.
1. Where is definition of $T_\rho^\mu$?
1. How is this used in the experiments? The data seems to naturally live in $R^d$.

Finally, although I found the problem natural, the motivations given seem rather weak. Is memory such a big constraint? (The experiments are all modest datasets.) I didn't understand the discussion around "the disappearance of one subpopulation within a subset of measures may be difficult to detect...".

**Audience:**

Yes

**Audience Explanation:**

This would interest anyone working on optimal transport in the context of computation and learning. It's a clear fit for TMLR.

I'm not convinced what practical problems this will help on (more below), but even still the question is quite natural and worthy of acceptance.

**Broader Impact Concerns:**

None.

**Claims And Evidence:**

Yes

**Claims Explanation:**

Proposition 3.1 assumes that the mean measure has a support size larger than $K$. But later results apply the proposition in settings where the distribution is fixed and we let $K\to \infty$. I assume this is not a fundamental issue, but it needs to be fixed.

---------------------------
Update (6/18/26)

This issue was indeed easy to resolve, and has been fixed in the revision. I updated my "Claims" answer to "yes."

**Requested Changes:**

Main items:
1. Fix the issues around Prop 3.1 and its applications.
1. Modify/clarify the motivation discussion.
1. Improve the discussion around Linearized Optimal Transport

A couple small suggestions:
1. I was unfamiliar with the $\Sigma_K$ notation for simplex, consider defining it the first time you use it.
1. Some of the equation referencing is weird and renders as "Problem equation 1 yields...". Also, capitalize "equation," I think?

---

> ### Author Response · Authors · 2026-06-16
>
> Thank you for your feedback and insightful comments. We address your questions below.
>
> **About the assumption of Proposition 3.1**
> > We thank the reviewer for pointing out this inconsistency. We have revisited the proof of Proposition 3.1, and noticed that the restriction to $F_K$, i.e. requiring the centers to be distinct, is in fact unnecessary. The constructions of the modified Voronoi cells in Appendix A remain valid when centers $(x_k)_{1\leq k\leq K}$, are not pairwise distinct. In this case, cells might be empty.
>
> >We have therefore removed the restriction to $F_K$. This consequently eliminates the assumption on the cardinality of the mean measure and resolves the inconsistency with the asymptotic regime $K\rightarrow \infty$.
>
> **About the discussion on Linearized Optimal Transport**
> > We moved the introduction of Brenier's theorem and the map $T_{\rho}^{\mu}$ to the Background section for clarity. We state more explicitely that, when the measure $\rho$ is absolutely continuous, $T_{\rho}^{\mu}$ denotes the unique OT map pushing $\rho$ to $\mu$.
>
> > Regarding the experiments, our data indeed consist of point clouds in $\mathbb{R}^d$. More precisely, each observation is *not* a single point in $\mathbb{R}^d$, but a probability measure represented by a set of points. Since standard ML tasks are designed for vector data rather than collections of probability measures, we first embed each measure into a Hilbert space using the LOT embedding. The downstream learning tasks are then performed on the embeddings.
>
> **About the motivation discussion**
> > For datasets such as those arising in flow cytometry, one usually considers several patients, each represented by a measure composed of more than $10^5$ points, lying in a space of dimension greater than $10$. Consequently, simultaneously computing LOT embeddings for all measures across the dataset becomes computationally expensive. The main challenge is not memory constraint, but rather the efficiently computing the LOT embeddings.
>
> > Regarding the toy experiment on rare populations, we consider three probability measures, each defined as a mixture of the same 5 Gaussian distributions. The only difference between these three measures lies in their mixture weights, which are vectors in the 5-dimensional simplex $\Sigma_5$. We then draw samples from these probability measures. One of the component, shown in blue in Figure 2, represents the rare population. In the first measure (Group 1), it has a standard prevalence, in the second measure (Group 2), it becomes rare, and in the third measure (Group 3), it is completely absent. The experiment illustrates the benefit of mean-measure quantization in that specific case. By jointly considering the three measures to determine a common set of centroids and then approximating each population using these shared centroids, we can detect the vanishing population, originating from the blue Gaussian component. This has been clarified in the revision.
>
> **Small suggestions (simplex introduction and equation rendering)**
> > We thank the reviewer for pointing this out. We have corrected these details in the revision.

---

> > ### Comment · Reviewer_yA2a · 2026-06-18
> >
> > Thank you, this addresses all of my concerns. I will update my review.

---

### Review · Reviewer_YaUj · 2026-07-20

**Summary Of Contributions:**

The paper studies mean-measure quantization (MMQ), which is a method for approximating a collection of $N$ probability measures $\\mu^{(1)}, \\ldots, \\mu^{(N)}$ by discrete measures sharing a common $K$-point support. Proposition 3.1 shows this shared-support quantization problem is exactly equivalent to a single $K$-point quantization of the mean measure $\\bar\\mu = \\frac{1}{N}\\sum_{i=1}^N \\mu^{(i)}$. The paper establishes consistency (Theorem 3.3) and propagates it to convergence guarantees for downstream OT-based quantities: Wasserstein barycenters, statistical dispersion, clustering variance, and covariance operators of linearized OT (LOT) and kernel mean embeddings. Experiments on synthetic Gaussian data, a rare-population synthetic example, flow cytometry data, and satellite imagery compare MMQ (full and subsampled) against individual quantization, random subsampling, and empirical measures.

Strengths:
- Proposition 3.1's equivalence is correctly proven and elegant .
- The downstream convergence results in Section 4 cover a useful range of quantities.
- The paper is explicit (Remark 5.1) about the gap between the consistency theory, which applies to the full mean measure, and the practically deployed Subsampled MMQ, and explicit (Remark 4.7) about the restrictiveness of the L-Lipschitz Monge map assumption underlying Proposition 4.4. Both are honest disclosures of scope.
- Experimental design is reasonably varied and the real-data comparisons against individual quantization (a strong baseline) are honest.

Weaknesses:
- The dimension-dependent rates (Proposition 3.5, Proposition 4.4) are acknowledged qualitatively to suffer from the curse of dimensionality, but their actual numeric magnitude at the experimental $K, d$ used (e.g., $d=10$ for flow cytometry) is not reported, leaving unclear whether the guarantees are informative or vacuous on the paper's own showcase dataset.
- Proposition 4.1  is qualitative only, with no rate, unlike the other Section 4 results.
- No significance testing accompanies the real-data accuracy comparisons (Figures 3a, 6a), so "MMQ is comparable to IQ" is a visual claim.

**Audience:**

Yes

**Audience Explanation:**

Scalable OT-based learning from collections of measures is an area with concrete applications (flow cytometry, image collections), and both the theoretical framework and the empirical case for mean-measure subsampling would interest that audience.

**Claims And Evidence:**

Yes

**Claims Explanation:**

Proposition 3.1 and Theorem 3.3 are correctly proven, and the downstream propositions in Section 4 follow soundly from them. The authors are transparent about two important scope limitations: the consistency theory is proven for quantization of the full mean measure, while the computationally attractive variant used in the experiments is Subsampled MMQ, with the added approximation error explicitly flagged as outside the current analysis (Remark 5.1); and the L-Lipschitz assumption on Monge maps underlying Proposition 4.4 is acknowledged as restrictive (Remark 4.7).

My one concern is that the curse-of-dimensionality caveat, while acknowledged qualitatively in the conclusion, is not quantified. Given that the flow cytometry experiment operates at $d=10$ and the bounds in Proposition 3.5 and Proposition 4.4 involve rates like $K^{-2/d}$ (and a further quarter-power for the covariance bound), it would strengthen the paper to report what these bounds actually evaluate to at the $K$ range used ($10$--$200$).

**Requested Changes:**

1. Please report the numeric value of the bounds in Proposition 3.5 and/or Proposition 4.4 at the $K$ and $d$ actually used in the flow cytometry experiment, to let readers judge whether the guarantees are quantitatively informative in that setting rather than only asymptotically true.

2. Please add a rate, or at least brief discussion of expected convergence speed, for Proposition 4.1 (barycenter convergence), for consistency with the other Section 4 results.

3. Please add error bars or a simple significance test to the real-data classification comparisons (Figures 3a, 6a).

4. Consider adding a brief caveat in the main text (Section 5.1.2) that the interpretability benefit illustrated by the rare-population experiment relies on the rare subpopulation being sufficiently represented in at least some of the measures; if it is negligible across all measures, standard k-means quantization of the pooled mean measure may absorb it entirely.